**Subject Category:**
Biology (whole organism)

environmental science

common-pool resources, ecosystem services, natural resources, sustainable use, wetlands

**Author for correspondence:**
Falko T. Buschke
e-mail: falko.buschke@gmail.com

# Ecosystem services and ecological degradation of communal wetlands in a South African biodiversity hotspot

## A. Owethu Pantshwa and Falko T. Buschke

Centre for Environmental Management (67), University of the Free State, P.O. Box 339 Bloemfontein 9300, South Africa

FTB, 0000-0003-1167-7810

Wetlands provide important ecosystem services to rural communities. However, wetlands are often on communal land, so they may become degraded when individual users act to maximize their personal benefit from ecosystem services without bearing the full environmental costs of their actions. Although it is possible to manage communal resources sustainably, this depends on the dynamics of the socio-ecological system. In this study, we used a structured questionnaire to examine whether demographic characteristics of a rural community and the propensity for partaking in damage-causing activities affected the benefits obtained from the wetlands. Responses from 50 households in the rural Hlabathi administrative area within the Maputo-Albany-Pondoland Biodiversity Hotspot, South Africa, indicated that the entire community obtains some benefits from wetlands; most notably regulating ecosystem services. However, males were more likely to benefit from wetlands, which highlights a potential power imbalance. Respondents were more likely to blame others for wetland degradation, although there was no link between the damage-causing activities and benefits from wetlands. The high dependence on ecosystem services by community members, when combined with gender-based power imbalances and the propensity to blame others, could jeopardize the sustainable use of communal wetlands. Therefore, we describe how strong leadership could nurture a sustainable social–ecological system by integrating ecological information and social empowerment into a multi-level governance system.

# 1. Introduction

Wetlands provide many ecosystem services to rural communities [1,2]. These rural communities often lack the adequate built infrastructure to supply water, so they rely directly on wetlands for their drinking and domestic water. Since wetlands are often their sole source of water, rural communities also use these habitats to supply water and grazing for livestock and, in many instances, for subsistence cultivation [3]. These activities put pressure on wetland ecosystems and jeopardize their ecological integrity and long-term functioning.

In addition to direct benefits, rural communities also obtain indirect benefits from wetlands that often go unnoticed [4,5]. Indirect benefits include water purification, sediment retention and flood attenuation [4,6]. These indirect benefits are often overlooked by communities, which can exacerbate wetland degradation. When communities are unaware of the indirect benefits of wetlands, they may view these ecosystems as unused wastelands to be transformed for commercial or residential development [7]. This suggests that people might degrade wetlands without realizing that they are also cutting off their own supply of ecosystem services.

Another issue, particularly in Africa, is that wetlands regularly occur on communal land and become common-pool resources. Common-pool resources are resources that are rival (i.e. using the resources reduces their availability to other users), but non-excludable (i.e. no single individual can restrict others from accessing these resources) [2]. This often leads to the tragedy of the commons [8], whereby individuals maximize their personal benefit from common resources without bearing the full costs of their actions. Instead, these costs are shared among all the users of the resource.

It is difficult, though not impossible, to solve the tragedy of the commons. One solution is collective-choice rules [9], where users of wetlands design and enforce their own rules on how to share the resources. However, this solution only works where community members who rely on wetlands are also the ones damaging the wetlands. If, for example, the poor and uneducated depend on wetlands the most, but a different group of wealthy individuals graze or plough these ecosystems, then a power imbalance exists. Such power dynamics may threaten the sustainable use of common-pool resources [10,11].

This study was carried out in the Hlabathi area of the Eastern Cape Province, South Africa. This rural area is characterized by traditional settlements and communal land-use for subsistence agriculture but is also in the heart of the globally significant conservation area: Maputo-Albany-Pondoland Biodiversity Hotspot. Communities here rely on wetlands for their drinking water and share these ecosystems with livestock and small-scale cultivation. However, as is often the case with common-pool resources, no one seems to take responsibility for the ecological functioning of these ecosystems.

We used a questionnaire to evaluate whether the local community at Hlabathi benefited from the supply of wetland ecosystem services. There are suggestions that the benefits of some ecosystem services, like food production, outweigh the associated declines of other services [12]. So, we assessed whether respondents benefited more from provisioning ecosystem services (water, plants for medicine or food), regulating ecosystem services (flood attenuation, sediment retention) or cultural ecosystem services (expression of traditional culture). The benefits gained from wetlands were then compared with how often community members took part in activities that damage wetlands.

Demographic data complemented this information on benefits and impacts. These data were used to disaggregate how ecosystem services are used by different segments of the community [13,14]. Social capital (i.e. well-established networks of social bonds and norms) is known to promote sustainable resource use [9,15,16]. We assumed that older respondents and respondents who have lived in the region longer had more time to develop social capital. By contrast, power imbalances can inhibit the sustainable use of common-pool resources [11,17], so we evaluated whether power imbalances related to gender dynamics and hierarchical leadership influenced the way people used wetlands. Lastly, people with a larger stake in the sustainable use of natural resources have greater incentives to prevent ecosystem degradation [9,18]. We incorporated dependence on wetlands in our analyses by assuming that the uneducated and unemployed have fewer alternative sources of income and would, therefore, rely more on ecosystem services.

We analysed the information on the benefits from, and impacts to, wetlands in conjunction with demographic data. This allowed us to test the following two research questions: First, are the people in the Hlabathi region who benefit the most from wetlands the same set of people who are responsible for most of the activities that damage these wetlands? This was dissected further to identify which types of ecosystem service were most beneficial to respondents and whether respondents acknowledged that their own actions could lead to wetland degradation. The second research question was whether demographic characteristics—such as social capital, power dynamics and dependence on natural

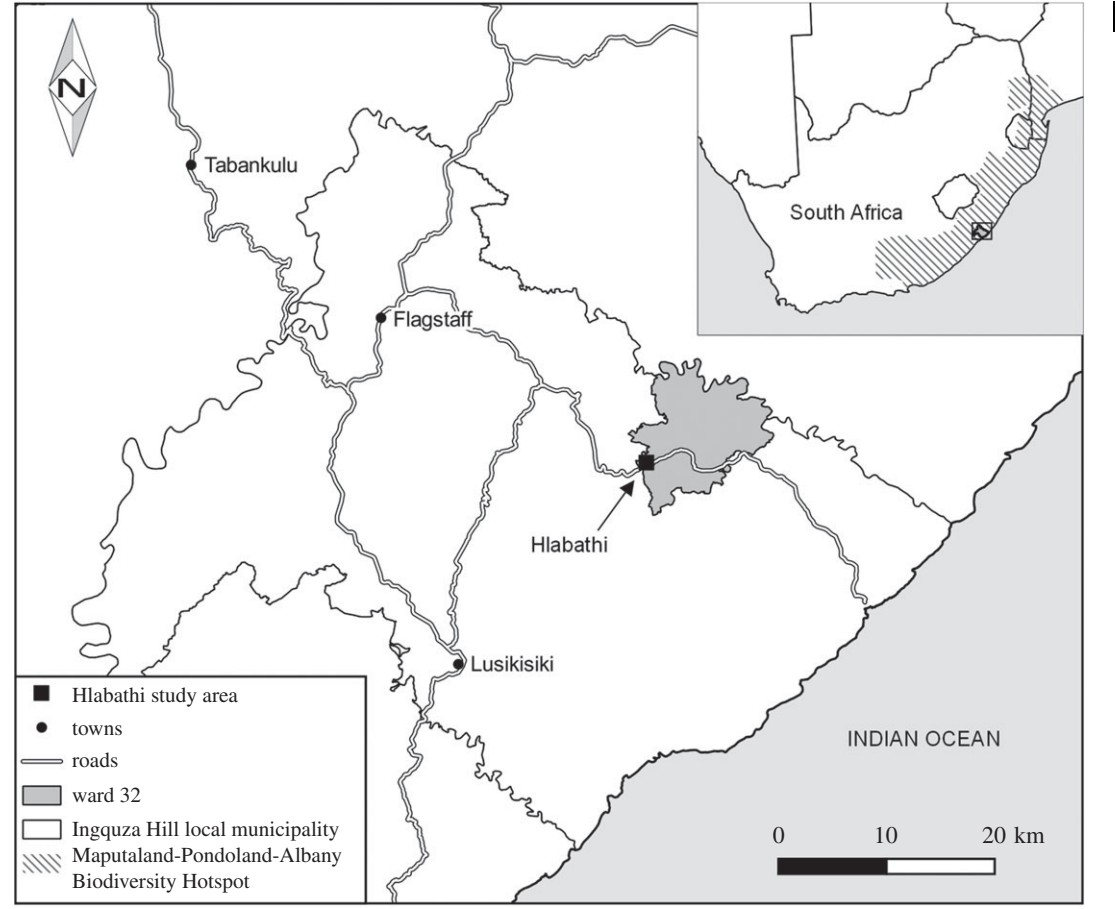

**Figure 1.** The geographical position of the Hlabathi administrative area.

resources—explained the relationship between wetland benefits and impacts. Answering these two questions provided the first step towards the sustainable utilization of the Hlabathi wetlands as a common-pool resource.

# 2. Material and methods

## 2.1. Study area

This study took place in the Hlabathi administrative area located along the Wild Coast of the Eastern Cape, South Africa (figure 1). This area comprises approximately 150 households located in ward 32 of the Ingquza Hill local municipality. The ward is situated within the environmentally sensitive Maputo-Albany-Pondoland (MAP) Biodiversity Hotspot [19,20]. As an internationally recognized biodiversity hotspot, MAP is regarded as a geographical area where the local loss of biodiversity would have a global impact [21].

The Hlabathi administrative area contains more than 200 wetlands of various hydrogeomorphic units [22]. These wetlands are not within protected areas (<7% of MAP is formally protected) and because there are few restrictions to using these wetlands they are prone to degradation by human activities. The Integrated Development Plan (IDP) of the Ingquza Hill local municipality [23] indicates that these wetlands are used for a variety of traditional community activities. Moreover, 71.8% of the municipal area lacks access to a piped water supply, implying that many people rely on wetlands for their daily water usage.

According to the IDP [23], the local community has low levels of education. Only 2.4% of the population had completed secondary school and just 1.4% of the population had post-secondary qualifications. Functional literacy was at 48% within the municipal area. Furthermore, each of the approximately 150 households was made up of five to six people, for which the total unemployment was 56% and youth unemployment was 61%.

## 2.2. Survey design

Community surveys were conducted with all community members, as well as their leadership (both traditional leaders and ward councillors) during July and August 2016. The process was facilitated by collaboration with municipal officials and support from municipal resources. Due to low literacy levels, the questionnaire was administered orally by trained fieldworkers. Moreover, the questionnaire had to be translated into Xhosa to ensure that the respondents understood the questions. Historically, Xhosa was the predominant language of the region, but there was additional cultural homogenization during the second half of the twentieth century when the Apartheid government established the Transkei homeland as a quasi-autonomous ethnic nation state. This created a language barrier, which meant that the questionnaire had to be kept as simple as possible to minimize potential translation or interpretation errors. The survey group comprised one respondent from each of the 50 randomly selected households. Since there were approximately 150 households in the region, this survey represented 33% of the total households. Prior to the survey, community members were informed of the purpose of the research, they were ensured that their participation was voluntary and that their individual responses would be kept anonymous.

## 2.3. Questionnaire design

The questionnaire was designed to measure the following: (i) the benefits obtained from wetlands, (ii) how often respondents partook in the most common damage-causing activities to wetlands (which were clearly visible during an earlier reconnaissance visit), and (iii) respondents' demographic variables. Data from the questionnaire were analysed using R version 3.3.1 [24].

Section A of the questionnaire covered respondents' demographic information, including the gender, language, age, level of education, occupation, duration of residency and community role. Section B comprised 16 questions about the interaction of respondents with wetlands. In Section B of the questionnaire, odd-numbered questions were used to determine if the respondent benefited from wetlands. These eight questions about benefits from wetlands represented three types of ecosystem services: provisioning (Q1 and Q7 water; Q3 and Q15 plants), regulating (Q5 floods and erosion; Q11 insect pests) and cultural services (Q9 and Q13 expression of culture). Differences between these categories were tested using a non-parametric Friedman rank sum test for blocked data (`friedman.test` command in `stats` package). This was followed by two *post hoc* multiple comparisons pairwise tests, which both reported the same results: a Conover test with Bonferroni correction for multiple comparisons (`posthoc.friedman.conover.test` command in `PMCMR` package) and a Nemenyi test (`posthoc.friedman.nemenyi.test` command in `PMCMR` package). These two tests can be viewed as non-parametric equivalents to paired *t*-test and Tukey tests, respectively.

The even-numbered questions in Section B of the questionnaire were used to determine damage-causing activities (grazing, ploughing, dumping and burning). Although ploughing and dumping waste are clearly detrimental to wetland integrity, historical grazing by wild ungulates, contemporary grazing by livestock and natural burning regimes can be important drivers of wetland functioning [25,26]. However, lack of rotational grazing and inadequate fire management in the region has led to chronic degradation of the wider landscape [27,28]. Because respondents may have been biased in their estimation of their damage to wetlands, we only asked four questions related to damage-causing activities, but duplicated these to quantify the respondents' own experience as well as their perceived experiences of others in their community (giving a total of eight questions). This is a simplified version of the normative technique, which weighs up responses relative to what are perceived as societal or cultural norms [29]. Self-reported benefits and damages were compared with those perceived for others in the community using a non-parametric Wilcoxon test (`wilcox.test` command in `stats` package). Self-reported and perceived community responses were also compared using dependent sample assessment plots (`granovagg.ds` command in `granovaGG` package).

For each odd-numbered question, responses were coded based on whether the respondent *Never* (0), *Occasionally* (1) or *Very Often* (2) benefited from wetlands. One question (Q11), however, described the ecosystem disservice of wetlands hosting mosquitoes, flies and other pests. For this question, the coding was reversed. Even-numbered questions were designed to determine whether respondents took part in activities that were potentially damaging to wetlands. Responses were also coded based on whether the respondent *Never* (0), *Occasionally* (1) or *Very Often* (2) participated in damage-causing activities. For each respondent, the benefit from, and damage to, wetlands were quantified as their mean responses to odd- and even-number questions, respectively.

We examined odd- and even-numbered responses to ensure that they represented unidimensional scales for the benefits obtained from wetlands and damage-causing activities, respectively. To do this, we tested for internal consistency and dimensionality by calculating Cronbach's alpha ($\alpha$), McDonald's hierarchical omega ($\omega_h$) and explained common variance (ECV) (omega command in psych package). Alpha and omega values greater than 0.7 and ECV values greater than 0.6 suggest that responses represent a unidimensional scale and can, therefore, be aggregated into a single dimension [30,31]. These analyses identified multidimensionality in the questionnaire, so the questions that represented different scales were removed from subsequent analyses to ensure that aggregated responses represented single scales. Five questions (Q1, Q3, Q11, Q13, Q15) were aggregated to represent a single scale for the benefits obtained from wetlands ($\alpha = 0.761$, $\omega_h = 0.713$ and ECV = 0.683). Similarly, five questions (Q4, Q6, Q10, Q12, Q14) could be aggregated to represent a single scale for damage-causing activities on wetlands ($\alpha = 0.832$, $\omega_h = 0.693$ and ECV = 0.634).

We evaluated the effect of seven demographic variables on the benefits obtained from wetlands. These included 'age' and 'duration of residence' in the village, which were continuous variables coded as the mid-point of the age ranges specified in the questionnaire (table 1). 'Language' was recorded as a categorical variable, but this information was discarded after all but two respondents indicated they were Xhosa, meaning that there was insufficient language variation within the community for subsequent analyses (the two other respondents were English and seSotho). 'Gender', 'education', 'community role' and 'employment' were included as categorical variables (table 1).

The benefit obtained from wetlands was modelled using a multiple linear regression model (lm command in the stats package) with the mean value for damage-causing responses and the six demographic variables as explanatory variables. Collinearity among predictors was not a concern because the variance inflation factors (1.56–2.64) were all lower than the recommended threshold of 4. Statistical significance of these explanatory variables was assessed at an alpha significance level of 0.05 using an $F$-test associated with a Type I analysis of variance (anova command in stats package). Although ordinary least-squares regression is not always ideal for regressing one survey-based scale on another, we favoured it over alternative approaches (e.g. major-axis regression), which cannot be used to evaluate multiple explanatory variables simultaneously.

## 3. Results

Of the 50 respondents in this study, 58% were female, 42% male, and most were in their 20s or 30s (table 1). The respondents tended to live in the area for more than 16 years, but roughly a third of the respondents only immigrated to the area in the last year. There were clear indications of high unemployment rates and low levels of education (table 1).

There was no discernible statistical difference between self-reported benefits from ecosystem services and the perceived benefits to other members in the community (Wilcoxon $V = 602$, $p = 0.752$; figure 2), which implies that community members perceive their benefits to be in line with those of the rest of the community. All forms of benefits accrued more than occasionally, with regulating ecosystem services perceived as being more beneficial than provisioning or cultural ecosystem services ($\chi^2 = 8.91$, $p = 0.011$; figure 3). By contrast, even though there was alignment between self-reported damage-causing activities and the perceived damage caused by others in the community, respondents were more likely to attribute damage to others (Wilcoxon $V = 158$, $p < 0.001$; figure 2).

The multiple regression model showed that male respondents were more likely to benefit from wetland ecosystem services (table 2), which is indicative of potential gender-based power imbalances in the region. However, no other demographic variable was statistically associated with the benefits derived from wetlands. This implies that, with the exception of gender, the community in Hlabathi was relatively homogeneous in terms of how they benefited from wetlands. Predictor variables of social capital (age and duration of residency), power dynamics (community role) and dependency (education and employment) did not influence the way community members benefited from wetlands (table 2). Notably, respondents who were more likely to partake in grazing, ploughing, dumping or burning did not stand to benefit any more from wetland ecosystem services than their peers who partook in these activities less frequently.

## 4. Discussion

This study showed that all community members benefited from ecosystem services from wetlands in the Hlabathi administrative area, South Africa, but men tended to benefit more than women. Besides this

**Table 1.** Summary of the demographic variables for the 50 respondents in the Hlabathi administrative area.

| variable | no. respondents | proportion |
|---|---|---|
| **gender** | | |
| female | 29 | 0.58 |
| male | 21 | 0.42 |
| **age** | | |
| younger than 18 | 1 | 0.02 |
| 18 – 29 years | 22 | 0.44 |
| 30 – 39 years | 10 | 0.20 |
| 40 – 49 years | 8 | 0.16 |
| 50 – 59 years | 7 | 0.14 |
| older than 60 | 2 | 0.04 |
| **duration of residence** | | |
| Less than 1 year | 16 | 0.32 |
| 2 – 5 years | 1 | 0.02 |
| 6 – 15 years | 0 | 0.0 |
| 16 – 25 years | 18 | 0.36 |
| more than 30 years | 15 | 0.30 |
| **community role** | | |
| community member | 34 | 0.68 |
| student | 4 | 0.08 |
| other | 12 | 0.24 |
| **employment** | | |
| community leader | 2 | 0.04 |
| employed | 6 | 0.12 |
| unemployed | 40 | 0.80 |
| pensioner | 2 | 0.04 |
| **education** | | |
| primary education | 10 | 0.20 |
| secondary education | 20 | 0.40 |
| tertiary education | 2 | 0.04 |
| other (including some secondary school) | 18 | 0.36 |

gender difference, which could reflect an underlying power imbalance, the use of ecosystem services was consistent regardless of social capital, power dynamics and dependence on natural resources. Moreover, there was no evidence that the men and women who degraded wetlands gained more from wetland ecosystem services than their fellow community members. This has both positive and negative consequences for the sustainable management of wetlands in the region.

Wetlands are often regarded as a free resource used for public good [5] and, in many instances, they are not privately owned and exist on communal land where everyone has the freedom to exploit them. However, people have different needs and interests and may affect the natural environment in different ways [8]. Free access to natural resources by people only serving their own ends often leads to environmental degradation [8]. In the case of the Hlabathi wetlands, there was no formal community control on how these ecosystems are used. However, communal resources need not necessarily be degraded. Ostrom [9] synthesized the variables that are known to support the sustainable use of common-pool resources. These included aspects related to (i) the resources themselves, (ii) the social characteristics of the user community, and (iii) the governance systems in place.

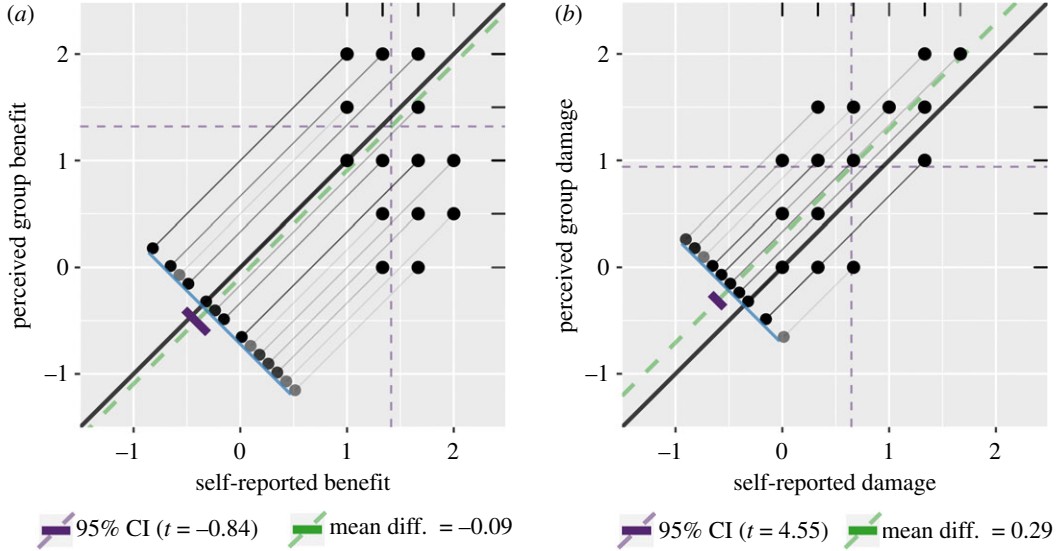

**Figure 2.** Dependent sample assessment plots comparing the self-reported and perceived group benefits (*a*) from and activities damaging (*b*) to wetlands. The solid black identity line represents perfect concordance, the dashed green line is the mean difference between the two sets of scores (with the solid purple bar as the 95% confidence interval) and the dashed purple lines denoting the mean values for each score.

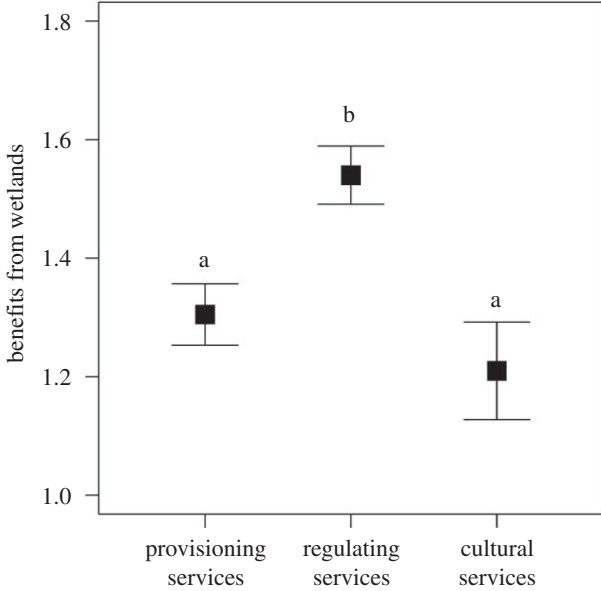

**Figure 3.** The benefits from wetlands reported by respondents for three categories of ecosystems services. The vertical axis is how often respondents benefited from wetlands (0 = Never, 1 = Occasionally, 2 = Very often). Letters denote significantly different groups from the *post hoc* multiple comparisons pairwise tests.

## 4.1. The nature of communal wetlands

Communal resources are more likely to be used sustainably if they are geographically immobile, small in size, highly productive and display predictable dynamics [9]. Although this study at the Hlabathi wetlands did not focus explicitly on these biophysical features, some generalizations are possible. Wetlands are immobile, which makes their sustainable utilization more likely than other communal resources, like seasonal grazing lands or migratory species, which require spatially dynamic usage rules during different times of the year (e.g. [32]). Similarly, although there are many discrete wetlands in the Hlabathi administrative area, each only covers a small geographical area, which should also promote sustainable use. However, the productive and predictable supply of ecosystem services from wetlands might be potential obstacles to their sustainable use. Communities tend not to self-organize when natural resources are very abundant or highly depleted [9,33]. If degradation reduces the productivity

**Table 2.** Analysis of variance table from the multiple regression model between self-reported benefits from wetlands as a dependent variable and how often respondents took part in damage-causing activities and demographic variables as independent variables (* Denotes coefficients that differ significantly from 0, and NS denotes coefficients that do not differ significantly from 0 at an alpha significance level of 0.05)

| variable | coefficient (s.e.) | F-score | d.f. | p-value |
|---|---|---|---|---|
| **intercept** | 1.506 (0.26) | | | |
| **damage-causing activities** | −0.034 (0.08) | 0.30 | 1 | 0.585[NS] |
| **age** | −0.004 (<0.01) | 0.14 | 1 | 0.709[NS] |
| **education** (secondary school as baseline) | | 0.04 | 3 | 0.866[NS] |
| primary school qualification | 0.051 (0.10) | | | |
| tertiary qualification | −0.245 (0.18) | | | |
| other qualification | −0.031 (0.09) | | | |
| **gender** (female as baseline) | | 8.79 | 1 | 0.005* |
| male | 0.221 (0.08) | | | |
| **employment** (community leader as baseline) | | 0.475 | 3 | 0.701[NS] |
| employed | −0.010 (0.20) | | | |
| unemployed | −0.183 (0.21) | | | |
| pensioner | 0.0081 (0.26) | | | |
| **community role** (community member as baseline) | | 3.163 | 2 | 0.054[NS] |
| student | −0.121 (0.16) | | | |
| other | −0.192 (0.10) | | | |
| **duration of residence** | 0.007 (<0.01) | 3.29 | 1 | 0.078[NS] |

s.e. = standard error; d.f. = degrees of freedom.

and predictability of the wetlands, then community members would be less inclined to act sustainably. This could trap communities in a cycle that makes sustainable collective action less probable.

The predictable supply of wetland services might be jeopardized by climate change [5], because the region is expected to experience hotter temperatures and more frequent extreme weather events [34]. This erodes self-organization and collective action because local community members will be less likely to follow rules when the future outcomes of their actions are uncertain [9,35]. Therefore, future strategies to manage the Hlabathi wetlands more sustainably should reframe the link between these ecosystems and local communities to better reflect the dynamic biophysical functioning and adaptive capacity of the system [36,37]. This will necessitate not only a better understanding of future wetland dynamics but also of the relationships between different community members and of their nature connectedness [38].

## 4.2. Social characteristics

There are clear opportunities to improve communal resource use at the Hlabathi wetlands. Ostrom [9] listed the five predictors of sustainable use as (i) a small number of users, (ii) a dependence on the common-pool resource, (iii) strong leadership, (iv) robust norms and social capital, and (v) mental models that incorporate socio-ecological sustainability. Since there were only 150 households in the Hlabathi administrative area and everyone seemed to be dependent on the wetlands in some form (figure 3 indicates that everyone benefits from wetlands more than occasionally), the first two social predictors already apply. Furthermore, community members seem to be aware that they rely on wetlands as much as their neighbours do (figure 2 shows concordance between self-reported and perceived community benefits). Community members have a shared interest in preserving wetland systems, which frees them from the conflict caused when different community members have different land-use expectations (e.g. [39,40]). Therefore, interventions should focus on developing strong leadership, social cohesion and an environmental knowledge-base to improve the sustainability of wetland use.

The results from the questionnaire show that men benefit more from Hlabathi wetlands than women. Gender is an important determinant of pro-environmental behaviour because women may favour

environmental altruism more than men [41]. Understanding how these different segments of the community relate to ecosystem services clarifies how natural resources are linked to social development [13,14]. Sustainable wetland utilization will be difficult unless all stakeholders in the community can contribute to decision-making and influence collective-choice rules [42]. A future research priority would be to understand whether gender differences in our study were merely the consequence of gender-differentiated environmental values and awareness, or whether they reflect deeper power imbalances and cultural taboos. For instance, Cocks and colleagues [43] reported gender-based restrictions in Xhosa culture, where certain activities in nature are limited to men (e.g. initiation ceremonies) or women (e.g. fuel wood collection). If this is also the case with wetland ecosystem services, then these deeply rooted cultural traditions will not be overcome through superficial community development programmes aiming to promote gender equity.

Another finding that needs to be addressed is that respondents were more likely to attribute blame for wetland destruction to members of the community other than themselves. There are two potential explanations for this: respondents could indeed partake in damage-causing activities less often than the rest of their community or they under-reported their own negative actions. There is cause for concern in the former instance, because people's behaviour tends to reflect what they perceive as social norms [44–46]. If this is the case at the Hlabathi wetlands, then it is an obvious obstacle to sustainable social norms and mental models. However, it is more likely that respondents under-reported their own activities. Respondents may have been displaying a social-desirability bias, which is a systematic error caused by trying to project a favourable image of themselves relative to the rest of society [47]. Alternatively, they could be blaming others as a mechanism to cope with an environmental situation seemingly beyond their control [48].

Regardless of the reason for this discrepancy in responses, it illustrates the complex interplay between the psychology of rural communities and their natural surroundings. Initiatives that aim to use wetlands more sustainably will fail if they ignore the underlying psychology at play. A deeper understanding of individual psychology needs to be translated into collective action and embedded in formal and informal institutions [49]. This is where strong conservation leadership is necessary. Such leadership can be encouraged by combining the strengths of multiple individuals that share the vision and values of the rest of the community [50,51].

## 4.3. Governance systems

The final predictor of sustainable use of common-pool resources is good governance structures with collective-choice rules [9]. These rules would describe how grazing and burning best practice (e.g. [25,26]) should be implemented by local users. Defining such rules would arise from an action arena for negotiations between government and other informal institutions [52], where the focus is less on *which* management actions should be used and more on *how* these actions should be implemented. Therefore, sustainable governance in the region is an issue of coordination and collective actions, in addition to overcoming the potential intrinsic discrepancies in the power dynamics between males and females [10,11].

Presently, it seems that the common-pool wetlands are being over-exploited because individuals maximize their own utility while externalizing negative effects to the community. This is not because respondents are generally selfish or have negative perceptions of nature. On the contrary, respondents were Xhosa, a cultural group that has been shown previously to score highly on various measures of nature connectedness [31,43]. Such human–biosphere connectedness is central to the sustainable use of ecosystems and resilient social–ecological systems [53,54]. The challenge, therefore, lies in translating individual nature connectedness to the broader society and its institutions [49].

It is possible that communicating scientific information to traditional leadership structures could establish sustainable practices throughout the wider community. For example, McCartney and colleagues [55] reported that local chiefs perceived wetlands more positively after their study on wetland hydrology and ecosystem services in Ga-Mampa, South Africa. Influential community members, like traditional leaders, change the way information flows through society, which is a known lever for promoting sustainability [56]. However, leadership—whether formal or traditional—cannot command sustainable behaviour in a top-down fashion. Instead, it needs to be embedded into a multi-level governance structure (e.g. [57]) that includes national initiatives, like the Working for Wetlands expanded public works programme that stemmed from the successful Working for Water programme [58]; and grassroots actions by individuals who help define the rules and structure of the broader social–ecological system [56]. The latter consideration should be supported by encouraging individuals that their actions make a difference to societal norms and standards of the community as a whole [49,59].

## 4.4. Roadmap to sustainable communal wetlands

This study showcases the complex and somewhat counterintuitive ways rural communities interact with natural resources. Even though individuals benefit from the services supplied by wetlands, they still partake in activities that jeopardize the very systems they depend on. Understanding this dynamic lies at the heart of using wetlands more sustainably. Although South African decision-makers and stakeholders generally acknowledge that social dynamics and public education are important for sustainable usage of water resources, they tend to attribute lower priority to these issues compared with technical solutions, like built infrastructure and environmental regulation [60].

Therefore, we recommend that future initiatives promote social issues to equal standing with technical considerations. Specifically, building on our results, we make three recommendations on how the Hlabathi wetlands can be used more effectively based on the interplay between ecology, society and institutions. First, the ecological dynamics of these ecosystems should be quantified and this information should be shared with the community, possibly through traditional leadership structures. Second, strong leadership (both formal and informal) could empower community members to define their own collective future. This is likely to require a reframing of social norms and standards and overcoming gender-based power imbalances. Third, collectively defined social rules should be integrated into a multi-level governance system, where individuals, traditional leaders and government officials collaborate towards the shared goal of sustainable natural resource management.

Ethics. All respondents provided inform consent prior to participation. This study was approved by both the Ingquza Hill Municipality and the University of the Free State.

Data accessibility. The data and R-scripts are available on the figshare online data repository [61].

Authors' contributions. Both authors designed the questionnaire and wrote the manuscript, A.O.P. carried out the surveys and F.T.B. analysed the data.

Competing interests. We declare we have no competing interests.

Funding. We received no funding for this study.

Acknowledgements. We thank the respondents, fieldworkers and staff at the Ingquza Hill local municipality for supporting this survey. Nancy Job, Mark Difford, Ruchi Badola and anonymous reviewers provided useful comments that improved this manuscript.

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
