## [Reviewer comments · Royal Society Open Science]

Review History

RSOS-171030.R0 (Original submission)

Review form: Reviewer 1 (Mark Difford)

Is the manuscript scientifically sound in its present form?

No

Are the interpretations and conclusions justified by the results?

No

Is the language acceptable?

Yes

Is it clear how to access all supporting data?

No

Do you have any ethical concerns with this paper?

No

Have you any concerns about statistical analyses in this paper?

Yes

Recommendation?

Major revision is needed (please make suggestions in comments)

Comments to the Author(s)

Overview:---

This is a relatively well written paper that presents the results of a study of wetland-use by a community living in an ecologically sensitive part of South Africa. The dataset on which the results are based is valuable, and the results of the study are likely to be of some interest even to a general audience. Publication of the paper in its present form cannot however be recommended because (1) inappropriate methods have been used for some of the analyses; (2) the results of some of the analyses are incorrectly interpreted; and (3) the scales that have been used to measure benefit from and damage to wetlands are insufficiently unidimensional, and to some extent are conflated (they partly measure each other). Once these problems have been fixed, and parts of the paper rethought, I think that the results would be worth publishing.

Below I briefly review the main problems I identified, beginning with a set of lesser problems. In the last part of my review, under the head Detailed Comments, I address these and other issues in greater detail and more directly, on a line-by-line basis. I have also attached a report in which I provide further proofs of most of the key issues and code to support the alternative analyses that I propose.

Relatively Minor, but Nonetheless Important Problems Are:---

1) The authors treat the individual items of each scale, i.e. the 16 questions of their survey, as if they are interval- or ratio-level variables, i.e. continuous variables. However, each question is an ordinal-level variable (an ordered factor) with just three categories (Never, Sometimes, Always). This affects their calculation of statistics on the reliability, validity, and dimensionality of each scale, where Pearson correlations are used. Polychoric correlations should have been used instead. Revelle's omega function, which the authors used to calculate these statistics, uses Pearson correlations by default, but there is an argument to that function that allows polychoric correlations to be used (in the event that scale items have fewer than ~ seven categories or levels). Incorrectly treating individual items as if they are continuous variables also affects the results reported in Table 3, where the means and standard errors of individual questions are calculated and reported. In my Detailed Comments I propose a replacement analysis, which I flesh out in my attached report. Good options are to use Wilcoxon signed-rank tests or Cliff's delta statistics (R package orddom). See Carifio & Perla (2007) and Brown (2011) for notes on the distinction between the items of a scale and the scale itself, and what the acceptable numerical treatment of each is. Also see Cliff (1993, 1996).

2) The authors fit no-intercept regressions (but do not tell the reader); the results are reported in Table 2. They then make the mistake of interpreting the statistics that result from those regressions as if they derive from regressions that were fitted with an intercept term. R^2 (coefficient of determination) values from intercept- and no-intercept regressions have fundamentally different interpretations (true for all reputable statistical software programs). Notably, the R^2 value from a no-intercept regression does not represent the amount of variance of the regressand (response) that the regressor (predictor) explains; it also does not represent the extent to which the regressand and the regressor are associated. This is relatively easy to demonstrate. The Pearson correlation coefficient (measures linear association) between the regressor and the regressand of the first regression listed in Table 2 (Benefits from wetlands) is 0.140. If we square this we get the R^2 for an ordinary least-squares model fitted with an intercept term, which is 0.020, compared to 0.944 for the no-intercept regression. Given that the

accepted interpretation of models fitted with an intercept term represents the proportion of variance in the response variable that the predictor explains (here ~ 2% compared to ~ 94% for the no-intercept regression), R^2 values from no-intercept models cannot, ipso facto, have the same interpretation but must have a wholly different one. The R^2 value from a no-intercept regression in fact measures the proportion of the variability *_about the origin_* that the regression explains, and not the proportion of variability of the response variable as a whole, or around the grand mean, that the regression explains. See Hocking (1996) and Eisenhauer (2003) and elsewhere.

Additionally, the estimated slope-coefficients of no-intercept models depend on the strong theoretical assumption that the expected value of the response variable is zero when the value of the predictor variable is zero. The authors provide no evidence to support such a claim, which, indeed, seems to be part of what they are trying to prove. For the first regression reported in Table 2, the slope coefficient for an intercept-based regression is 0.10 not 0.88. It might also be noted that the estimates reported for the first regression (Benefits from wetlands) are in any case wrong because the authors mistakenly included two questions, viz Q5 and Q11, in both the regressor and the regressand when calculating the mean-scores of sub-scales that they use in the regression.

While it might be of some interest to know how the two subscales of each scale relate to each other (i.e. how the self-assessed and perceived-group effects relate to each other), this type of analysis should not be necessary if the main scales are sharpened (or refined) to be truly unidimensional. The authors would probably be better off carrying out some form of agreement/concordance analysis, though Lin's concordance correlation coefficient (Lin 1989; Lin et al. 2002) is probably too exacting. Bland-Altman (i.e. Tukey's mean-difference) plots, coupled with paired t-tests or a non-parametric or Bayesian alternative, would probably be adequate. See the R packages `granovaGG` (function `granovagg.ds`), `epiTools`, `MethComp`, `Agreement`, amongst others.

This brings me to a final issue under this head. Ordinary least-squares regression is not appropriate when both the regressor and the regressand are measured with approximately the same degree of error, as they are here. The methods I have just referred to, viz agreement/concordance analysis, use major-axis (or Deming) regression as a central tool. It is the appropriate type of regression to use when regressing one psychometric or survey-based scale on another one. See R packages `lmodel2` and `smatr` and references therein.

3) The authors use pairwise correlations to assess the collinearity of putative predictors for the model they summarize in Figure 3 and Table 4. Pairwise correlations are neither an appropriate nor a recommended way of assessing collinearity, and do not provide grounds for excluding variables from the model. The reason of this is that collinearity depends on jointly assessing all the predictors in the model. Recommended approaches rely on (generalized) variance-inflation factors and condition indices of the design matrix (Belsley et al. 1980 [2004]). See `vif()` in Fox's `car` package for R, and `ols_coll_diag()` in Hebbali's `olsrr` package for R.

Were the authors to use these methods to test for collinearity they would find that neither of the two variables they exclude come close to showing signs of being collinear with other variables. In fact, the most collinear putative predictor is Position (role played in the community), which has a generalized variance-inflation factor of 2.64. This is well below the recommended value of >4 , which is when one begins to suspect that there might be a sufficient degree of collinearity to begin to affect parameter estimates. Values above 10 are generally recognized as indicating a degree of collinearity that needs fixing.

The Main Problem:---

The authors results and conclusions rest on the reliability, unidimensionality, and validity of the

two scales they use. Having explored these scales in considerable detail while preparing this review I am not convinced by the authors' argument that the scales are sufficiently unidimensional for mean-scores (that have a clear interpretation) to be formed, and therefore for the analysis to proceed.

In supporting their argument, the authors cite Wilhelm-Rechmann et al. (2014) [which I was a coauthor of] and Reise et al. (2010). However, there are fundamental differences between the arguments of Wilhelm-Rechmann et al. (2014) and the present study, notably: (1) the statistics on unidimensionality reported by Wilhelm-Rechmann et al. are better than those reported here; (2) the scale Wilhelm-Rechmann et al. used, the NEP (New Environmental Paradigm) scale, is a known quantity that has been used in hundreds of studies worldwide with reliable results, despite showing signs of being weakly multidimensional; (3) five-point items, which the NEP scale uses, are more likely to give the spurious impression of multidimensionality than three-point items are; and (4) Wilhelm-Rechmann et al. were able to demonstrate the robustness of their findings by favourably comparing them with (a) the results of a meta-analysis of 69 studies spanning 36 nations, and (b) the results of Dunlap et al.'s survey of residents of Washington State (USA). See Wilhelm-Rechmann et al. (2014) for details. The authors of the present paper unfortunately do not have access to such "back-stop" studies that might be used to sway the argument or corroborate their results.

In rejecting the authors' arguments on this fundamentally important question (how much multidimensionality is too much) I have additionally been swayed by the following facts. In deciding whether a multidimensional scale is sufficiently unidimensional for analysis to proceed safely, Reise (see the reference list and the help file for Revelle's omega function in R) stresses an index known as the explained common variance (ECV), which he calls a 'cleaner index of "unidimensionality"' than omega hierarchical (Reise et al. 2010). He and his associates recommend that this be > 0.70 , except if the percentage of uncontaminated correlations (PUC) is high (> 0.80), in which case an $ECV > 0.50$ would be acceptable (Reise et al. 2016). When PUC is less than 0.80 then Reise and associates recommend that $ECV > 0.60$ and that omega hierarchical be > 0.70 . Only then can one be reasonably sure that the presence of some multidimensionality is not severe enough to disqualify the interpretation of the scale as being primarily unidimensional (Reise et al. 2013).

The problem is that the authors' scales have ECV and PUC values of 0.38/0.36 (Pearson/polychoric correlations) and 0.64/0.55, respectively, for the benefits scale, and ECV and PUC values of 0.34/0.36 and 0.56/0.56, respectively, for the damage-focused scale. Revelle's beta coefficient for the two scales is also unacceptably low, being 0.26/0.20 and 0.25/0.07, respectively. At the very least, it should be > 0.5 , and preferably should be > 0.7 (Revelle 1979, Cooksey & Soutar 2006). In short, there is no evidence to support the authors' view that their scales are sufficiently unidimensional for them to be treated as if they are unidimensional, and hence for mean (or summated) scores to have a clear interpretation.

As they stand, the two scales are moderately correlated ($r = 0.378$). This indicates a degree of conflation between them, a suspicion that is confirmed by exploratory analysis using tools designed to select unidimensional scales. Mokken Scale Analysis (R package *mokken*), for instance, excludes Q5, Q11, and Q15 from any scale, and puts three items from the scale to assess damage to the wetland (Q2, Q8, Q16) on the scale to assess benefits from the wetland, and it puts one item from the scale to assess benefits from the wetland (Q9) on the scale to assess damage to the wetland. These Mokken-based scales, with mixed questions from the original two scales, have considerably better statistics on reliability and dimensionality than the scales formed by the authors; they also are not correlated ($r = 0.091$).

Using a set of tools that I routinely use, I was able to manually select items from each scale that

are indisputably unidimensional, based on the criteria suggested by Reise and associates (op. cit.). Significantly most of the items that I excluded, viz Q5, Q7, Q9, Q2, Q8, and Q16, are items that the foregoing exploratory methods either excluded or put on the other scale, viz Q9 on one hand, and Q2, Q8, and Q16 on the other. The two scales that are formed after these exclusions[*] are not correlated ($r = 0.072$) and have omega hierarchical, ECV, and PUC statistics (based on polychoric correlations) of 0.71, 0.68, and 0.67 for the benefit-scale, and 0.69, 0.63, and 0.53 for the damage-scale. These are good scale-statistics for data of this type.

[*] Revised benefits scale: [Q1,Q3,Q11,Q13,Q15] (excludes Q5,Q7,Q9); revised damage scale: [Q4,Q6,Q10,Q12,Q14] (excludes Q2,Q8,Q16).

The foregoing findings mean that were the authors prepared to revise their scales along the lines I have proposed, there is much that they could rescue from this interesting study (and worthwhile dataset) and be confident about the findings. There is no reason, I think, why at least some of the excluded items (i.e. questions), namely those that measure clearly defined activities like grazing (Q2) and burning (Q8, Q16), could not be used as ancillary variables in supporting analyses. For instance, they could perhaps be used as predictors in the replacement-analysis I suggest for Table 2. See the expository document/code that I have attached for ideas and supporting proofs.

References:---

Belsley, D. A.; Kuh, E. & Welsch, R. E., 1980 (paperback reprint 2004). *Regression Diagnostics: Identifying Influential Data and Sources of Collinearity*. John Wiley & Sons, Inc, .

Brown, J. D. (2011). Likert items and scales of measurement?, SHIKEN: JALT Testing & Evaluation SIG Newsletter 15 : 10-14.

Carifio, J. & Perla, R. J. (2007). Ten Common Misunderstandings, Misconceptions, Persistent Myths and Urban Legends about Likert Scales and Likert Response Formats and their Antidotes, *Journal of Social Sciences* 3 : 106-116.

Cliff, N. (1993). Dominance statistics: Ordinal analyses to answer ordinal questions, *Psychological Bulletin*, 1993, 114: 494-509.

Cliff, N. (1996). *Ordinal Methods for Behavioral Data Analysis*. Lawrence Erlbaum Associates, Inc.

Cooksey, R. W. & Soutar, G. N. (2006). Coefficient beta and hierarchical item clustering: An analytical procedure for establishing and displaying the dimensionality and homogeneity of summated scales, *Organizational Research Methods* 9 : 78-98.

Eisenhauer, J. G. (2003). *Regression through the Origin*, *Teaching Statistics* : 76-80.

Hocking, R. R., 2013. *Methods and Applications of Linear Models: Regression and the Analysis of Variance*. JOHN WILEY & SONS.

Lin, L.; Hedayat, A. S.; Sinha, B. & Yang, M. (2002). Statistical Methods in Assessing Agreement, *Journal of the American Statistical Association* 97 : 257-270.

Lin, L. I. (1989). A concordance correlation coefficient to evaluate reproducibility, *Biometrics* 45 : 255-268.

Reise, S. P.; Moore, T. M. & Haviland, M. G. (2010). Bifactor models and rotations: exploring the extent to which multidimensional data yield univocal scale scores, *J Pers Assess* 92 : 544-559.

Reise, S. P.; Scheines, R.; Widaman, K. F. & Haviland, M. G. (2013). Multidimensionality and Structural Coefficient Bias in Structural Equation Modeling: A Bifactor Perspective, *Educational and Psychological Measurement* 73 : 5-26.

Revelle, W. (1979). Hierarchical cluster analysis and the internal structure of tests, *Multivariate Behavioral Research* 14 : 57-74.

Wilhelm-Rechmann, A.; Cowling, R. M. & Difford, M. (2014). Responses of South African land-use planning stakeholders to the New Ecological Paradigm and the Inclusion of Nature in Self scales: Assessment of their potential as components of social assessments for conservation projects, *Biological Conservation* 180 : 206-213.

Detailed Comments:---

Page 2 Line 24: Strictly, "rival" should be "rivalrous." (Or, "resources that rival each other...but are non-excludable...." Or maybe, "...resources that are rivals...but are non-excludable....")

Page 2 Line 42--43: for [the] ecological functioning. Think you need to add the definite article.

Page 2 Line 43--50: The two "lists" in this paragraph are ambiguous. The "from" that starts the first list does not apply to all of its elements except if you itemize/enumerate them. If not, then you should use "from provisioning ecosystem services...from regulating ecosystem service...and from cultural ecosystem services...." More important is the second list, where there is an unclear contrast between the first three items and the "or" that separates the last item. Was this really an "either/or" quantification, or do you perhaps mean, "...the grazing, burning, and ploughing (cultivation) of wetlands, and whether..."? Surely all four activities were quantified?

Page 2 Line 51--52: First instance of "of" should be "on". For the latter, the complete statement would (and perhaps should) be, "information on the demographic data of respondents."

Page 2 Line 53: You need to state your rationale. "The reason for collecting demographic data is that...."

Page 3 Line 6: "respondents" needs a plural apostrophe (respondents') and a comma after "community". But the following might be better: "...the questionnaire asked respondents about their roles in the community, to distinguish...."

Page 3 Line 15: Should be, "The information on...."

Page 3 Line 34--36: Meaning of the last sentence is not clear; presumably it's that, because of the area's uniqueness, a loss of biodiversity would have a global impact. If so, then try, "..., MAP is regarded as a geographic[al]* area where a local loss of biodiversity would have a global impact." [* Note: Figure 1 uses "geographical" not "geographic".]

Page 4 Line 12--13: "..., which we assumed to be an underlying cause..." would be better.

Page 4 Line 24: Need "into Xhosa".

Page 4 Line 37: Possessive case: needs to be, "respondents' demographic variables...."

Page 4 Line 48--49: Incomplete. Try, "The even-numbered questions of Section B were used to determine [assess] damage-causing activities...."

Page 4 Para beginning Line 50--51: Based on an examination of your code, I don't see that your description matches what you did. For instance, for the benefits.self ~ benefits.group regression, the regressor and regressand share two questions (Q5, Q11), the other three being different (so five questions each); whereas for the damage.self ~ damage.group regression, the regressor and regressand are each based on four questions (so four questions each), with no shared questions. Clearly you made a mistake when calculating mean-scores for the benefits.self and benefits.group subscales. But even then, your description remains confusing and confused. It's clear to me, but only because I scrutinized your code and the accompanying analyses, that each scale is based on four questions, each of which was asked twice, giving eight sets of answers for each scale, and a total of 16 sets of answers for the two scales together. It is misleading to say, as you do, that

each question in section B was asked twice, because in the preceding paragraph (Line 41) you say that "Section B comprised of 16 questions." If each question was asked twice, and there are 16 questions in all, then there should be 32 sets of answers or scores, which there are not. Note also that sometimes you capitalize "Section B" in mid-sentence and sometimes you don't, writing "section B."

Page 4 Line 52--53: Leave out "asking."

Page 4 Line 54: Should be "are," not "is" [to see this, leave out "perceived"].

Page 4 Line 56 & Page 5 Line 6--9: See point 2 of my introductory remarks for discussion of the problems associated with these regressions and with your interpretation of them. Briefly, addressing just the core problem, you fitted no-intercept models, which fundamentally changes the interpretation of R^2 , the coefficient of determination. Notably, it loses all connexion to the correlation-based interpretation that you give it. And just as significantly, you omitted to tell the reader that this is what you did. The R^2 for the intercept-based benefits model is 0.01964, which is equal to $\text{cor}(y,x)^2$, which in turn is equal to a Pearson correlation of 0.1401. In short, this means that there is no demonstrable relationship between the two sets of scores, as measured by Pearson correlation (or, in fact, Spearman/Kendall correlations). Yet R^2 for the no-intercept version of the model is 0.944. An informed reader, unaware that you've fitted a model without an intercept (because nowhere do you say that that is what you fitted), would assume that a strong linear relationship between the two forms of the benefits variable exists (Pearson correlation = 0.972), when in fact this is not so. The same general remarks apply to the other regression. The results you present in Table 2, and in the second paragraph of Results on pages 5 and 6, are therefore not only wrong but highly misleading.

Page 5 Line 17: Should be "odd- and even-numbered...."

Page 5 Line 32--34: The following is not true: "...every single respondent was Xhosa, meaning that there was no language variation within the community." Your dataset shows that one respondent spoke seSotho and one spoke English. They may all have identified themselves as belonging to the Xhosa people, but that is an essentially different thing.

Page 5 Line 35--37 et seq.: Pairwise correlation is not a recommended/accepted method of testing for (multi)collinearity. See point 3 of my introductory remarks.

Page 5 Line 36: Should be "correlated with".

Page 5 Line 43--45: Results are selective because there are no grounds on the basis of collinearity for excluding the predictors you did. Also, you need to tell the reader that the F-values and P-values you report in Table IV are based on a sequential decomposition of the sums of squares, i.e. so-called Type I or sequential ANOVA, rather than on Type II or Type III (which here are the same).

Page 5 Line 51--52: "...and low levels of education..." is better (education is a noun, not an adjective).

Page 5 Line 52--55: The statistics you report are highly misleading and do not support your claims. As I've noted above, and in my introductory remarks, R^2 from a no-intercept regression does not measure association. To get that type of interpretation you need to fit a regression model with an intercept term. The square-root of the R^2 value from such a regression would indicate the degree of linear association and for your data is $r = 0.140$. The R^2 and beta (slope-coefficient) for the model are 0.020 ($r = 0.140 \approx 0.020^{.5}$) and 0.100, respectively.

Page 6 Line 39--43: This is the same issue as that just noted, namely interpreting the statistics from a no-intercept model as if it's an intercept-based model. Categorically, the R^2 value from a no-intercept model does not estimate the amount of variance in the response variable explained by the predictor(s). See point 2 of my introductory comments and the references I cite there. If one fits a model with an intercept term then the variance in self-reported damage that the model explains is 36% not 75%, and the slope-coefficient is 0.652 not 0.528. Hence, respondents are not twice as likely to attribute damage to someone else than to themselves. Pearson's correlation coefficient is 0.584.

Page 6 Table 2 Caption: Somewhat muddled, and you have written "from from" (repeated). Maybe try something along the lines of: "Summary statistics from separate linear regressions of self-reported benefits and self-reported damage on perceived community benefits and perceived community damage, respectively."

Page 6 Table 2: The statistics you report do not have the interpretation you have given them. As I have noted above, you interpret these statistics as if you fitted models with an intercept term but the truth is that you fitted no-intercept models while omitting to say so. Further, for the no-intercept model for Benefits, the slope-coefficient you report should be 0.883 not 0.880.

Page 6 Line 55--56: "...wetland, results found a positive, statistically significant relationship...." Results themselves do not find, they suggest or indicate some result or finding. You might say, "...wetland, we found..." or "...wetland, the results indicate..." or "...wetland, the results support the finding/fact..."

Page 7 Line 30--32: You fitted an additive model (Table 4, Figure 3), which conditions on, or takes account of, the effect of other predictors in the model; it does not test whether the effect of self-reported damage on self-reported benefits (Figure 3-a) depends on values (or levels) of these other predictors. Hence, you provide no evidence to support your claim that the positive effect of self-reported damage on self-reported benefits is consistent across age, gender, level of education, and employment category. To test this, you would need to have fitted a multiplicative model in which self-reported damage interacted with the other predictors, i.e. a model of the form: benefit ~ damage * (other predictors). If you do this, then your claim indeed holds good; however, nowhere do you explicitly fit, discuss, or present the results of such a model.

Page 7 Line 34: Residency is not collinear if one uses a reliable test to test collinearity.

Page 7 Line 36--38 & Table 3: I am not convinced by this approach, which strictly speaking is wrong. The main problem is that you are taking means and standard errors of individual questions as if each is a continuous variables, which it is not. Each is an ordinal-level variable. While most authorities consider it to be permissible to take the mean of several ordinal-level items/questions and to treat the result as a continuous variable, this is not true for the analysis of individual questions. Each of your questions is an ordered factor with just three levels or categories (0,1, and 2), i.e. your questions are not an interval- or ratio-scale variables, which would allow the calculation of such statistics without criticism. The problem is essentially the same as one I mentioned above: when using Revelle's omega function to calculate statistics on reliability and dimensionality for your scales (to measure benefits and damage) you used Pearson correlation (the default) rather than polychoric correlations. See Carifio & Perla (2007), Brown (2011), and elsewhere on this issue. I also don't think that you have made it clear enough to the reader that these four activities represent the measurement variables of the scale you used to measure damage to the wetland, i.e. the eight even-numbered questions of your survey. Finally, if you were to use this type of approach (but you should not) then you need to use confidence intervals (bootstrapped) not standard errors. See §3 of my report where I suggest replacement analyses. For the Wilcoxon signed-rank tests that I carry out there, all you would need are the estimate of the location shift (Est. = pseudomedian), the associated 95% confidence intervals, and perhaps p-values (for exact p-values in the presence of ties see `wilcoxsign_test()` in package `coin`). Arguably a more powerful/detailed approach would be to use Cliff's delta statistics. See R package `orddom`.

Page 7 Line 36--38: To make this clear, you need to add "self-reported damage" or "Damage-causing activities" in brackets after "...they partook grazing, ploughing, dumping or burning" so that there is an explicit link to Table 4/Figure 3-a.

Page 7 Line 38: Preposition missing: usually one "partakes of" or "partook in." But the real problem here is that it is not the residents/respondents themselves who are doing the grazing, it's the livestock. You are looking for something simple like, "...was how often they used the wetland for grazing, ploughing, dumping or burning."

Page 7 §4. Discussion: In light of my foregoing criticisms on Method and Interpretation, much of

the Discussion will need to be rewritten. I have therefore reserved comment on this part of the manuscript for this round of reviews.

Page 8 Line 34 (Table 4): See note on this above, where I review Methods (Page 5 Line 43--45).

Review form: Reviewer 2

Is the manuscript scientifically sound in its present form?

Yes

Are the interpretations and conclusions justified by the results?

Yes

Is the language acceptable?

Yes

Is it clear how to access all supporting data?

Not Applicable

Do you have any ethical concerns with this paper?

No

Have you any concerns about statistical analyses in this paper?

No

Recommendation?

Accept as is

Comments to the Author(s)

Nicely written paper but small sample size. Large sample size (>350) may affect your result/inference.

Review form: Reviewer 3 (Ruchi Badola)

Is the manuscript scientifically sound in its present form?

No

Are the interpretations and conclusions justified by the results?

Yes

Is the language acceptable?

No

Is it clear how to access all supporting data?

Yes

Do you have any ethical concerns with this paper?

No

Have you any concerns about statistical analyses in this paper?

Yes

Recommendation?

Major revision is needed (please make suggestions in comments)

Comments to the Author(s)

The study is important as it is from a region with limited literature on community dependence on wetlands, thus it is an area of interest. However there are certain flaws that cannot be ignored. Hence this manuscript cannot be published in its present form.

Detailed comments are as:

1. The manuscript needs proper editing.
2. Introduction is fluid and lacks conviction
3. The title reads as "Ecosystem services and ecological degradation of communal wetlands in a South African biodiversity hotspot", but the study reported in the manuscript is from a single village located in the biodiversity hotspot and the study also reflects the results from a single wetland, hence there is a complete mis-match of the study design and the title.
4. The study area description fails to provide the overview on the area and type of wetland and the ecosystem services provided therein by the wetland. It is crucial to enumerate few of the ecosystem services and goods provided by the particular wetland, which is the focus of the study.
5. It will be useful to know the species assemblage in Ward 32 where the study has been conducted. Particularly interesting will be the name of some endemic species of the region
6. Page 4; Lines 6-8: The authors have mentioned Protected Areas only once in the entire manuscript. However it will be interesting to know the percentage coverage of the MAP under protected area network.
7. Page 4; lines 9-10: It will be useful to provide a list of traditional activities carried out by people in the wetlands.
8. What kind of environment degradation the authors are referring to? The authors need to specify.
9. Page 4; Lines 15-17: "Furthermore, each of the 56% and the youth unemployment ratio was 61%."

Replace ratio with percent.

10. Page 4: Questionnaire Design

This section needs to be re-written to reflect the variables used, the type of questions used (open-ended or close-ended). If close ended what was scale used?

"Differences between these categories were tested using a repeated measures ANOVA (lme

command in nlme package) followed by a post-hoc multiple comparisons pairwise t-test with Bonferroni correction for multiple comparison”

Do the author mean that difference between responses to the ecosystem services were tested? The authors need to rewrite this section to make the design clear.

Reading the results and specifically Figure 2 show that the authors have tried the t-test between the two groups of respondents: one which always derived benefits and the other group which occasionally derives benefits. This analysis doesn't make any sense. What is the scale used in figure 2 (Y-Axis)? It is unclear.

The methodology is flawed and the authors need to rework on the explanation of questionnaire design and analysis. I suggest that they refer to some previously published papers on wetland benefits and community attitude (Ambastha et al., 2008; Hussain and Badola, 2008, 2009; Badola et al. 2013).

11. Page 9; lines 35-39: The authors have justly used the Ostrom's concept of sustainable use in the discussion but have failed to reflect the same in their methods and results.

My overall impression is that the MS needs major revision.

Ruchi Badola

Wildlife Institute of India, Dehradun

Decision letter (RSOS-171030.R0)

26-Jan-2018

Dear Dr Buschke:

Manuscript ID RSOS-171030 entitled "Ecosystem services and ecological degradation of communal wetlands in a South African biodiversity hotspot" which you submitted to Royal Society Open Science, has been reviewed. The comments from reviewers are included at the bottom of this letter.

In view of the criticisms of the reviewers, the manuscript has been rejected in its current form. However, a new manuscript may be submitted which takes into consideration these comments.

Please note that resubmitting your manuscript does not guarantee eventual acceptance, and that your resubmission will be subject to peer review before a decision is made.

Your resubmitted manuscript should be submitted by 26-Jul-2018. If you are unable to submit by this date please contact the Editorial Office.

Please note that Royal Society Open Science will introduce article processing charges for all new submissions received from 1 January 2018. Charges will also apply to papers transferred to Royal Society Open Science from other Royal Society Publishing journals, as well as papers submitted as part of our collaboration with the Royal Society of Chemistry (<http://rsos.royalsocietypublishing.org/chemistry>). If your manuscript is submitted and accepted for publication after 1 Jan 2018, you will be asked to pay the article processing charge, unless you request a waiver and this is approved by Royal Society Publishing. You can find out more about the charges at <http://rsos.royalsocietypublishing.org/page/charges>. Should you have any queries, please contact openscience@royalsociety.org.

on behalf of Kevin Padian (Subject Editor)
openscience@royalsociety.org

Subject Editor Comments to Author:

One reviewer was positive but offered no substantial comments. The other two reviewers found severe problems which in their views would prevent publication of the manuscript. We will allow a resubmission, if the authors choose, but only if all the criticisms of the referees are adequately assessed, which may take considerable additional work. Thanks for submitting here and good luck.

Reviewers' Comments to Author:

Reviewer: 1

Comments to the Author(s)

Overview:---

This is a relatively well written paper that presents the results of a study of wetland-use by a community living in an ecologically sensitive part of South Africa. The dataset on which the results are based is valuable, and the results of the study are likely to be of some interest even to a general audience. Publication of the paper in its present form cannot however be recommended because (1) inappropriate methods have been used for some of the analyses; (2) the results of some of the analyses are incorrectly interpreted; and (3) the scales that have been used to measure benefit from and damage to wetlands are insufficiently unidimensional, and to some extent are conflated (they partly measure each other). Once these problems have been fixed, and parts of the paper rethought, I think that the results would be worth publishing.

Below I briefly review the main problems I identified, beginning with a set of lesser problems. In the last part of my review, under the head Detailed Comments, I address these and other issues in greater detail and more directly, on a line-by-line basis. I have also attached a report in which I provide further proofs of most of the key issues and code to support the alternative analyses that I propose.

Relatively Minor, but Nonetheless Important Problems Are:---

1) The authors treat the individual items of each scale, i.e. the 16 questions of their survey, as if they are interval- or ratio-level variables, i.e. continuous variables. However, each question is an ordinal-level variable (an ordered factor) with just three categories (Never, Sometimes, Always). This affects their calculation of statistics on the reliability, validity, and dimensionality of each scale, where Pearson correlations are used. Polychoric correlations should have been used instead. Revelle's omega function, which the authors used to calculate these statistics, uses Pearson correlations by default, but there is an argument to that function that allows polychoric correlations to be used (in the event that scale items have fewer than ~ seven categories or levels). Incorrectly treating individual items as if they are continuous variables also affects the results reported in Table 3, where the means and standard errors of individual questions are calculated and reported. In my Detailed Comments I propose a replacement analysis, which I flesh out in my attached report. Good options are to use Wilcoxon signed-rank tests or Cliff's delta statistics (R package *orddom*). See Carifio & Perla (2007) and Brown (2011) for notes on the distinction between the items of a scale and the scale itself, and what the acceptable numerical treatment of each is. Also see Cliff (1993, 1996).

2) The authors fit no-intercept regressions (but do not tell the reader); the results are reported in Table 2. They then make the mistake of interpreting the statistics that result from those regressions as if they derive from regressions that were fitted with an intercept term. R^2 (coefficient of determination) values from intercept- and no-intercept regressions have fundamentally different interpretations (true for all reputable statistical software programs). Notably, the R^2 value from a no-intercept regression does not represent the amount of variance of the regressand (response) that the regressor (predictor) explains; it also does not represent the extent to which the regressand and the regressor are associated. This is relatively easy to demonstrate. The Pearson correlation coefficient (measures linear association) between the regressor and the regressand of the first regression listed in Table 2 (Benefits from wetlands) is 0.140. If we square this we get the R^2 for an ordinary least-squares model fitted with an intercept term, which is 0.020, compared to 0.944 for the no-intercept regression. Given that the accepted interpretation of models fitted with an intercept term represents the proportion of variance in the response variable that the predictor explains (here ~ 2% compared to ~ 94% for the no-intercept regression), R^2 values from no-intercept models cannot, ipso facto, have the same interpretation but must have a wholly different one. The R^2 value from a no-intercept regression in fact measures the proportion of the variability about the origin that the regression explains, and not the proportion of variability of the response variable as a whole, or around the grand mean, that the regression explains. See Hocking (1996) and Eisenhauer (2003) and elsewhere.

Additionally, the estimated slope-coefficients of no-intercept models depend on the strong theoretical assumption that the expected value of the response variable is zero when the value of the predictor variable is zero. The authors provide no evidence to support such a claim, which, indeed, seems to be part of what they are trying to prove. For the first regression reported in Table 2, the slope coefficient for an intercept-based regression is 0.10 not 0.88. It might also be noted that the estimates reported for the first regression (Benefits from wetlands) are in any case wrong because the authors mistakenly included two questions, viz Q5 and Q11, in both the regressor and the regressand when calculating the mean-scores of sub-scales that they use in the regression.

While it might be of some interest to know how the two subscales of each scale relate to each other (i.e. how the self-assessed and perceived-group effects relate to each other), this type of analysis should not be necessary if the main scales are sharpened (or refined) to be truly unidimensional. The authors would probably be better off carrying out some form of agreement/concordance analysis, though Lin's concordance correlation coefficient (Lin 1989; Lin

et al. 2002) is probably too exacting. Bland-Altman (i.e. Tukey's mean-difference) plots, coupled with paired t-tests or a non-parametric or Bayesian alternative, would probably be adequate. See the R packages `granovaGG` (function `granovagg.ds`), `epiTools`, `MethComp`, `Agreement`, amongst others.

This brings me to a final issue under this head. Ordinary least-squares regression is not appropriate when both the regressor and the regressand are measured with approximately the same degree of error, as they are here. The methods I have just referred to, viz agreement/concordance analysis, use major-axis (or Deming) regression as a central tool. It is the appropriate type of regression to use when regressing one psychometric or survey-based scale on another one. See R packages `lmodel2` and `smatr` and references therein.

3) The authors use pairwise correlations to assess the collinearity of putative predictors for the model they summarize in Figure 3 and Table 4. Pairwise correlations are neither an appropriate nor a recommended way of assessing collinearity, and do not provide grounds for excluding variables from the model. The reason of this is that collinearity depends on jointly assessing all the predictors in the model. Recommended approaches rely on (generalized) variance-inflation factors and condition indices of the design matrix (Belsley et al. 1980 [2004]). See `vif()` in Fox's R package for R, and `ols_coll_diag()` in Hebbali's `olsrr` package for R.

Were the authors to use these methods to test for collinearity they would find that neither of the two variables they exclude come close to showing signs of being collinear with other variables. In fact, the most collinear putative predictor is Position (role played in the community), which has a generalized variance-inflation factor of 2.64. This is well below the recommended value of >4 , which is when one begins to suspect that there might be a sufficient degree of collinearity to begin to affect parameter estimates. Values above 10 are generally recognized as indicating a degree of collinearity that needs fixing.

The Main Problem:---

The authors results and conclusions rest on the reliability, unidimensionality, and validity of the two scales they use. Having explored these scales in considerable detail while preparing this review I am not convinced by the authors' argument that the scales are sufficiently unidimensional for mean-scores (that have a clear interpretation) to be formed, and therefore for the analysis to proceed.

In supporting their argument, the authors cite Wilhelm-Rechmann et al. (2014) [which I was a coauthor of] and Reise et al. (2010). However, there are fundamental differences between the arguments of Wilhelm-Rechmann et al. (2014) and the present study, notably: (1) the statistics on unidimensionality reported by Wilhelm-Rechmann et al. are better than those reported here; (2) the scale Wilhelm-Rechmann et al. used, the NEP (New Environmental Paradigm) scale, is a known quantity that has been used in hundreds of studies worldwide with reliable results, despite showing signs of being weakly multidimensional; (3) five-point items, which the NEP scale uses, are more likely to give the spurious impression of multidimensionality than three-point items are; and (4) Wilhelm-Rechmann et al. were able to demonstrate the robustness of their findings by favourably comparing them with (a) the results of a meta-analysis of 69 studies spanning 36 nations, and (b) the results of Dunlap et al.'s survey of residents of Washington State (USA). See Wilhelm-Rechmann et al. (2014) for details. The authors of the present paper unfortunately do not have access to such "back-stop" studies that might be used to sway the argument or corroborate their results.

In rejecting the authors' arguments on this fundamentally important question (how much multidimensionality is too much) I have additionally been swayed by the following facts. In deciding whether a multidimensional scale is sufficiently unidimensional for analysis to proceed

safely, Reise (see the reference list and the help file for Revelle's omega function in R) stresses an index known as the explained common variance (ECV), which he calls a 'cleaner index of "undimensionality"' than omega hierarchical (Reise et al. 2010). He and his associates recommend that this be > 0.70 , except if the percentage of uncontaminated correlations (PUC) is high (> 0.80), in which case an $ECV > 0.50$ would be acceptable (Reise et al. 2016). When PUC is less than 0.80 then Reise and associates recommend that $ECV > 0.60$ and that omega hierarchical be > 0.70 . Only then can one be reasonably sure that the presence of some multidimensionality is not severe enough to disqualify the interpretation of the scale as being primarily unidimensional (Reise et al. 2013).

The problem is that the authors' scales have ECV and PUC values of 0.38/0.36 (Pearson/polychoric correlations) and 0.64/0.55, respectively, for the benefits scale, and ECV and PUC values of 0.34/0.36 and 0.56/0.56, respectively, for the damage-focused scale. Revelle's beta coefficient for the two scales is also unacceptably low, being 0.26/0.20 and 0.25/0.07, respectively. At the very least, it should be > 0.5 , and preferably should be > 0.7 (Revelle 1979, Cooksey & Soutar 2006). In short, there is no evidence to support the authors' view that their scales are sufficiently unidimensional for them to be treated as if they are unidimensional, and hence for mean (or summated) scores to have a clear interpretation.

As they stand, the two scales are moderately correlated ($r = 0.378$). This indicates a degree of conflation between them, a suspicion that is confirmed by exploratory analysis using tools designed to select unidimensional scales. Mokken Scale Analysis (R package *mokken*), for instance, excludes Q5, Q11, and Q15 from any scale, and puts three items from the scale to assess damage to the wetland (Q2, Q8, Q16) on the scale to assess benefits from the wetland, and it puts one item from the scale to assess benefits from the wetland (Q9) on the scale to assess damage to the wetland. These Mokken-based scales, with mixed questions from the original two scales, have considerably better statistics on reliability and dimensionality than the scales formed by the authors; they also are not correlated ($r = 0.091$).

Using a set of tools that I routinely use, I was able to manually select items from each scale that are indisputably unidimensional, based on the criteria suggested by Reise and associates (op. cit.). Significantly most of the items that I excluded, viz Q5, Q7, Q9, Q2, Q8, and Q16, are items that the foregoing exploratory methods either excluded or put on the other scale, viz Q9 on one hand, and Q2, Q8, and Q16 on the other. The two scales that are formed after these exclusions[*] are not correlated ($r = 0.072$) and have omega hierarchical, ECV, and PUC statistics (based on polychoric correlations) of 0.71, 0.68, and 0.67 for the benefit-scale, and 0.69, 0.63, and 0.53 for the damage-scale. These are good scale-statistics for data of this type.

[*] Revised benefits scale: [Q1,Q3,Q11,Q13,Q15] (excludes Q5,Q7,Q9); revised damage scale: [Q4,Q6,Q10,Q12,Q14] (excludes Q2,Q8,Q16).

The foregoing findings mean that were the authors prepared to revise their scales along the lines I have proposed, there is much that they could rescue from this interesting study (and worthwhile dataset) and be confident about the findings. There is no reason, I think, why at least some of the excluded items (i.e. questions), namely those that measure clearly defined activities like grazing (Q2) and burning (Q8, Q16), could not be used as ancillary variables in supporting analyses. For instance, they could perhaps be used as predictors in the replacement-analysis I suggest for Table 2. See the expository document/code that I have attached for ideas and supporting proofs.

References:---

Belsley, D. A.; Kuh, E. & Welsch, R. E., 1980 (paperback reprint 2004). *Regression Diagnostics: Identifying Influential Data and Sources of Collinearity*. John Wiley & Sons, Inc, .

- Brown, J. D. (2011). Likert items and scales of measurement?, SHIKEN: JALT Testing & Evaluation SIG Newsletter 15 : 10-14.
- Carifio, J. & Perla, R. J. (2007). Ten Common Misunderstandings, Misconceptions, Persistent Myths and Urban Legends about Likert Scales and Likert Response Formats and their Antidotes, *Journal of Social Sciences* 3 : 106-116.
- Cliff, N. (1993). Dominance statistics: Ordinal analyses to answer ordinal questions, *Psychological Bulletin*, 1993, 114: 494-509.
- Cliff, N. (1996). *Ordinal Methods for Behavioral Data Analysis*. Lawrence Erlbaum Associates, Inc.
- Cooksey, R. W. & Soutar, G. N. (2006). Coefficient beta and hierarchical item clustering: An analytical procedure for establishing and displaying the dimensionality and homogeneity of summated scales, *Organizational Research Methods* 9 : 78-98.
- Eisenhauer, J. G. (2003). Regression through the Origin, *Teaching Statistics* : 76-80.
- Hocking, R. R., 2013. *Methods and Applications of Linear Models: Regression and the Analysis of Variance*. JOHN WILEY & SONS.
- Lin, L.; Hedayat, A. S.; Sinha, B. & Yang, M. (2002). Statistical Methods in Assessing Agreement, *Journal of the American Statistical Association* 97 : 257-270.
- Lin, L. I. (1989). A concordance correlation coefficient to evaluate reproducibility, *Biometrics* 45 : 255-268.
- Reise, S. P.; Moore, T. M. & Haviland, M. G. (2010). Bifactor models and rotations: exploring the extent to which multidimensional data yield univocal scale scores, *J Pers Assess* 92 : 544-559.
- Reise, S. P.; Scheines, R.; Widaman, K. F. & Haviland, M. G. (2013). Multidimensionality and Structural Coefficient Bias in Structural Equation Modeling: A Bifactor Perspective, *Educational and Psychological Measurement* 73 : 5-26.
- Revelle, W. (1979). Hierarchical cluster analysis and the internal structure of tests, *Multivariate Behavioral Research* 14 : 57-74.
- Wilhelm-Rechmann, A.; Cowling, R. M. & Difford, M. (2014). Responses of South African land-use planning stakeholders to the New Ecological Paradigm and the Inclusion of Nature in Self scales: Assessment of their potential as components of social assessments for conservation projects, *Biological Conservation* 180 : 206-213.

Detailed Comments:---

Page 2 Line 24: Strictly, "rival" should be "rivalrous." (Or, "resources that rival each other...but are non-excludable...." Or maybe, "...resources that are rivals...but are non-excludable....")

Page 2 Line 42--43: for [the] ecological functioning. Think you need to add the definite article.

Page 2 Line 43--50: The two "lists" in this paragraph are ambiguous. The "from" that starts the first list does not apply to all of its elements except if you itemize/enumerate them. If not, then you should use "from provisioning ecosystem services...from regulating ecosystem service...and from cultural ecosystem services...." More important is the second list, where there is an unclear contrast between the first three items and the "or" that separates the last item. Was this really an

"either/or" quantification, or do you perhaps mean, "...the grazing, burning, and ploughing (cultivation) of wetlands, and whether..."? Surely all four activities were quantified?

Page 2 Line 51--52: First instance of "of" should be "on". For the latter, the complete statement would (and perhaps should) be, "information on the demographic data of respondents."

Page 2 Line 53: You need to state your rationale. "The reason for collecting demographic data is that...."

Page 3 Line 6: "respondents" needs a plural apostrophe (respondents') and a comma after "community". But the following might be better: "...the questionnaire asked respondents about their roles in the community, to distinguish...."

Page 3 Line 15: Should be, "The information on...."

Page 3 Line 34--36: Meaning of the last sentence is not clear; presumably it's that, because of the area's uniqueness, a loss of biodiversity would have a global impact. If so, then try, "..., MAP is regarded as a geographic[al]* area where a local loss of biodiversity would have a global impact." [* Note: Figure 1 uses "geographical" not "geographic".]

Page 4 Line 12--13: "..., which we assumed to be an underlying cause..." would be better.

Page 4 Line 24: Need "into Xhosa".

Page 4 Line 37: Possessive case: needs to be, "respondents' demographic variables...."

Page 4 Line 48--49: Incomplete. Try, "The even-numbered questions of Section B were used to determine [assess] damage-causing activities...."

Page 4 Para beginning Line 50--51: Based on an examination of your code, I don't see that your description matches what you did. For instance, for the `benefits.self ~ benefits.group` regression, the regressor and regressand share two questions (Q5, Q11), the other three being different (so five questions each); whereas for the `damage.self ~ damage.group` regression, the regressor and regressand are each based on four questions (so four questions each), with no shared questions. Clearly you made a mistake when calculating mean-scores for the `benefits.self` and `benefits.group` subscales. But even then, your description remains confusing and confused. It's clear to me, but only because I scrutinized your code and the accompanying analyses, that each scale is based on four questions, each of which was asked twice, giving eight sets of answers for each scale, and a total of 16 sets of answers for the two scales together. It is misleading to say, as you do, that each question in section B was asked twice, because in the preceding paragraph (Line 41) you say that "Section B comprised of 16 questions." If each question was asked twice, and there are 16 questions in all, then there should be 32 sets of answers or scores, which there are not. Note also that sometimes you capitalize "Section B" in mid-sentence and sometimes you don't, writing "section B."

Page 4 Line 52--53: Leave out "asking."

Page 4 Line 54: Should be "are," not "is" [to see this, leave out "perceived"].

Page 4 Line 56 & Page 5 Line 6--9: See point 2 of my introductory remarks for discussion of the problems associated with these regressions and with your interpretation of them. Briefly, addressing just the core problem, you fitted no-intercept models, which fundamentally changes the interpretation of R^2 , the coefficient of determination. Notably, it loses all connexion to the correlation-based interpretation that you give it. And just as significantly, you omitted to tell the reader that this is what you did. The R^2 for the intercept-based benefits model is 0.01964, which is equal to $\text{cor}(y,x)^2$, which in turn is equal to a Pearson correlation of 0.1401. In short, this means that there is no demonstrable relationship between the two sets of scores, as measured by Pearson correlation (or, in fact, Spearman/Kendall correlations). Yet R^2 for the no-intercept version of the model is 0.944. An informed reader, unaware that you've fitted a model without an intercept (because nowhere do you say that that is what you fitted), would assume that a strong linear relationship between the two forms of the benefits variable exists (Pearson correlation = 0.972), when in fact this is not so. The same general remarks apply to the other regression. The results you present in Table 2, and in the second paragraph of Results on pages 5 and 6, are therefore not only wrong but highly misleading.

Page 5 Line 17: Should be "odd- and even-numbered...."

Page 5 Line 32--34: The following is not true: "...every single respondent was Xhosa, meaning that there was no language variation within the community." Your dataset shows that one respondent spoke seSotho and one spoke English. They may all have identified themselves as belonging to the Xhosa people, but that is an essentially different thing.

Page 5 Line 35--37 et seq.: Pairwise correlation is not a recommended/accepted method of testing for (multi)collinearity. See point 3 of my introductory remarks.

Page 5 Line 36: Should be "correlated with".

Page 5 Line 43--45: Results are selective because there are no grounds on the basis of collinearity for excluding the predictors you did. Also, you need to tell the reader that the F-values and P-values you report in Table IV are based on a sequential decomposition of the sums of squares, i.e. so-called Type I or sequential ANOVA, rather than on Type II or Type III (which here are the same).

Page 5 Line 51--52: "...and low levels of education..." is better (education is a noun, not an adjective).

Page 5 Line 52--55: The statistics you report are highly misleading and do not support your claims. As I've noted above, and in my introductory remarks, R^2 from a no-intercept regression does not measure association. To get that type of interpretation you need to fit a regression model with an intercept term. The square-root of the R^2 value from such a regression would indicate the degree of linear association and for your data is $r = 0.140$. The R^2 and beta (slope-coefficient) for the model are 0.020 ($r = 0.140 \approx 0.020^{.5}$) and 0.100, respectively.

Page 6 Line 39--43: This is the same issue as that just noted, namely interpreting the statistics from a no-intercept model as if it's an intercept-based model. Categorically, the R^2 value from a no-intercept model does not estimate the amount of variance in the response variable explained by the predictor(s). See point 2 of my introductory comments and the references I cite there. If one fits a model with an intercept term then the variance in self-reported damage that the model explains is 36% not 75%, and the slope-coefficient is 0.652 not 0.528. Hence, respondents are not twice as likely to attribute damage to someone else than to themselves. Pearson's correlation coefficient is 0.584.

Page 6 Table 2 Caption: Somewhat muddled, and you have written "from from" (repeated).

Maybe try something along the lines of: "Summary statistics from separate linear regressions of self-reported benefits and self-reported damage on perceived community benefits and perceived community damage, respectively."

Page 6 Table 2: The statistics you report do not have the interpretation you have given them. As I have noted above, you interpret these statistics as if you fitted models with an intercept term but the truth is that you fitted no-intercept models while omitting to say so. Further, for the no-intercept model for Benefits, the slope-coefficient you report should be 0.883 not 0.880.

Page 6 Line 55--56: "...wetland, results found a positive, statistically significant relationship...." Results themselves do not find, they suggest or indicate some result or finding. You might say, "...wetland, we found..." or "...wetland, the results indicate..." or "...wetland, the results support the finding/fact...."

Page 7 Line 30--32: You fitted an additive model (Table 4, Figure 3), which conditions on, or takes account of, the effect of other predictors in the model; it does not test whether the effect of self-reported damage on self-reported benefits (Figure 3-a) depends on values (or levels) of these other predictors. Hence, you provide no evidence to support your claim that the positive effect of self-reported damage on self-reported benefits is consistent across age, gender, level of education, and employment category. To test this, you would need to have fitted a multiplicative model in which self-reported damage interacted with the other predictors, i.e. a model of the form: benefit \sim damage * (other predictors). If you do this, then your claim indeed holds good; however, nowhere do you explicitly fit, discuss, or present the results of such a model.

Page 7 Line 34: Residency is not collinear if one uses a reliable test to test collinearity.

Page 7 Line 36--38 & Table 3: I am not convinced by this approach, which strictly speaking is wrong. The main problem is that you are taking means and standard errors of individual questions as if each is a continuous variables, which it is not. Each is an ordinal-level variable. While most authorities consider it to be permissible to take the mean of several ordinal-level items/questions and to treat the result as a continuous variable, this is not true for the analysis of individual questions. Each of your questions is an ordered factor with just three levels or categories (0,1, and 2), i.e. your questions are not an interval- or ratio-scale variables, which would allow the calculation of such statistics without criticism. The problem is essentially the same as one I mentioned above: when using Revelle's omega function to calculate statistics on reliability and dimensionality for your scales (to measure benefits and damage) you used Pearson correlation (the default) rather than polychoric correlations. See Carifio & Perla (2007), Brown (2011), and elsewhere on this issue. I also don't think that you have made it clear enough to the reader that these four activities represent the measurement variables of the scale you used to measure damage to the wetland, i.e. the eight even-numbered questions of your survey. Finally, if you were to use this type of approach (but you should not) then you need to use confidence intervals (bootstrapped) not standard errors. See §3 of my report where I suggest replacement analyses. For the Wilcoxon signed-rank tests that I carry out there, all you would need are the estimate of the location shift (Est. = pseudomedian), the associated 95% confidence intervals, and perhaps p-values (for exact p-values in the presence of ties see `wilcoxsign_test()` in package `coin`). Arguably a more powerful/detailed approach would be to use Cliff's delta statistics. See R package `orddom`.

Page 7 Line 36--38: To make this clear, you need to add "self-reported damage" or "Damage-causing activities" in brackets after "...they partook grazing, ploughing, dumping or burning" so that there is an explicit link to Table 4/Figure 3-a.

Page 7 Line 38: Preposition missing: usually one "partakes of" or "partook in." But the real problem here is that it is not the residents/respondents themselves who are doing the grazing, it's the livestock. You are looking for something simple like, "...was how often they used the wetland for grazing, ploughing, dumping or burning."

Page 7 §4. Discussion: In light of my foregoing criticisms on Method and Interpretation, much of the Discussion will need to be rewritten. I have therefore reserved comment on this part of the manuscript for this round of reviews.

Page 8 Line 34 (Table 4): See note on this above, where I review Methods (Page 5 Line 43--45).

Reviewer: 2

Comments to the Author(s)

Nicely written paper but small sample size. Large sample size (>350) may affect your result/inference.

Reviewer: 3

Comments to the Author(s)

The study is important as it is from a region with limited literature on community dependence on wetlands, thus it is an area of interest. However there are certain flaws that cannot be ignored. Hence this manuscript cannot be published in its present form.

Detailed comments are as:

1. The manuscript needs proper editing.

2. Introduction is fluid and lacks conviction

3. The title reads as "Ecosystem services and ecological degradation of communal wetlands in a South African biodiversity hotspot", but the study reported in the manuscript is from a single village located in the biodiversity hotspot and the study also reflects the results from a single wetland, hence there is a complete mis-match of the study design and the title.

4. The study area description fails to provide the overview on the area and type of wetland and the ecosystem services provided therein by the wetland. It is crucial to enumerate few of the ecosystem services and goods provided by the particular wetland, which is the focus of the study.

5. It will be useful to know the species assemblage in Ward 32 where the study has been conducted. Particularly interesting will be the name of some endemic species of the region

6. Page 4; Lines 6-8: The authors have mentioned Protected Areas only once in the entire manuscript. However it will be interesting to know the percentage coverage of the MAP under protected area network.

7. Page 4; lines 9-10: It will be useful to provide a list of traditional activities carried out by people in the wetlands.

8. What kind of environment degradation the authors are referring to? The authors need to specify.

9. Page 4; Lines 15-17: "Furthermore, each of the 56% and the youth unemployment ratio was 61%."

Replace ratio with percent.

10. Page 4: Questionnaire Design

This section needs to be re-written to reflect the variables used, the type of questions used (open-ended or close-ended). If close ended what was scale used?

"Differences between these categories were tested using a repeated measures ANOVA (lme command in nlme package) followed by a post-hoc multiple comparisons pairwise t-test with Bonferroni correction for multiple comparison"

Do the author mean that difference between responses to the ecosystem services were tested? The authors need to rewrite this section to make the design clear.

Reading the results and specifically Figure 2 show that the authors have tried the t-test between the two groups of respondents: one which always derived benefits and the other group which occasionally derives benefits. This analysis doesn't make any sense. What is the scale used in figure 2 (Y-Axis)? It is unclear.

The methodology is flawed and the authors need to rework on the explanation of questionnaire design and analysis. I suggest that they refer to some previously published papers on wetland benefits and community attitude (Ambastha et al., 2008; Hussain and Badola, 2008, 2009; Badola et al. 2013).

11. Page 9; lines 35-39: The authors have justly used the Ostrom's concept of sustainable use in the discussion but have failed to reflect the same in their methods and results.

My overall impression is that the MS needs major revision.
Ruchi Badola
Wildlife Institute of India, Dehradun

Author's Response to Decision Letter for (RSOS-171030.R0)

See Appendix A.

RSOS-180869.R0

Review form: Reviewer 2

Is the manuscript scientifically sound in its present form?

No

Are the interpretations and conclusions justified by the results?

No

Is the language acceptable?

No

Is it clear how to access all supporting data?

No

Do you have any ethical concerns with this paper?

No

Have you any concerns about statistical analyses in this paper?

No

Recommendation?

Reject

Comments to the Author(s)

The specific comments to the manuscript are:

This manuscript is about the ecosystem services and ecological degradation of the communal wetlands, but nowhere in the manuscript the authors have provided the specific ecosystem services to the local community of the study area. The authors have framed this manuscript around the perceived benefits to self as well as other. But how did the authors choose just a few of the provisioning (plant and water), regulating (flood and erosion) and cultural services, as mentioned in the questionnaire design.

What was the reason that most of the respondents were in their 20s and 30s, and how many were

leaders (traditional or formal), as in survey design it is mentioned that community survey was conducted with community members and their leadership.

Is community leader a form of employment? I don't agree.

To know demographic composition, it is advised to refer to the census report for the area, which will give accurate information on the demographic structure as well as economic and employment status.

It will be useful to provide a list of traditional activities carried out by people in the wetlands, in the Study Area section.

Page 4; lines 47-48: "According to the IDP [22], the local community has low levels of education, which we assumed to be an underlying cause of the environmental degradation."

Low level of education doesn't necessarily always results in environment degradation. Moreover, education doesn't grant the person to be environmentally conscious.

Page 6; Line 48: ".....All forms of benefits were accrued more than occasionally....", delete 'were'

Page 6; lines 53-54: "....The multiple regression model showed that male respondents were more likely to benefit from wetland ecosystem services (Table 2),"

Table 2 shows damage causing activities, whereas the sentence says that male respondents were more likely to benefit from wetland ecosystem.

The language is lax and needs proper editing for grammatical and syntax errors.

Review form: Reviewer 4

Is the manuscript scientifically sound in its present form?

Yes

Are the interpretations and conclusions justified by the results?

Yes

Is the language acceptable?

Yes

Is it clear how to access all supporting data?

Yes

Do you have any ethical concerns with this paper?

No

Have you any concerns about statistical analyses in this paper?

I do not feel qualified to assess the statistics

Recommendation?

Major revision is needed (please make suggestions in comments)

Comments to the Author(s)

The manuscript studies human-nature interactions using as a case study the ecological systems of wetlands and a rural community living in a biodiversity hotspot in South Africa as a social system. Interactions are based on benefits derived from the wetlands (ecosystem services) and damage-causing activities affecting the wetlands (ecological degradation). It brings to light: i) the relationship or lack of relationship between damage-causing activities, and benefits, and ii) the demographical factors that influence above interactions. The insights gained through this study are valuable as they cover an understudied, yet vulnerable area. However, the paper would benefit from a restructuring of content around the two enounced research questions. The manuscript went through the first round of revisions, and I acknowledge a lot of work has been invested in the statistical analysis, both from reviewers and authors. A general check for the English language is also recommended. Below are some specific comments.

ABSTRACT

The abstract correctly reflects the content of the paper.

Consider clarifying or separating the following statements for the abstract: "Respondents were more likely to blame others for wetland degradation, although there was no link between the damage-causing activities and benefits from wetlands." Is this reflected as such in the Results section?

INTRODUCTION

Lines 6-7: it is not clear how the stated goal "This study, therefore, set out to examine whether the people responsible for degrading wetlands are the same set of people who would benefit the most from their conservation" derives from the previous introductory sentences. Maybe delete "therefore" or make the link between motivation and aim clearer. Also, it is less common that the aim is stated so soon in a flow of a manuscript. Consider moving information about the aim of this study in the last paragraph of the introduction.

Lines 30-37: parts of this paragraph rather belong to section 2. Methods (a) Study area

Lines 38-65: most parts of these paragraphs rather belong to section 2. Methods (b) and (c)

Line 71: I am not sure "social capital" is a demographic characteristic. Consider adding "whether demographic characteristics and related social factors such as social capital, ..."

General comment Introduction: Consider restructuring the introduction so that each paragraph conveys a (general) message, that only foresees elements of the Discussion without revealing the results. For example, as a suggestion: a paragraph on wetlands in general, a paragraph on benefits/ecosystem services from wetlands, a paragraph on threats to wetlands in a certain context, (research gap), a paragraph on the aim and research questions.

METHODS

(a) Study area

Move here related material from the introduction.

Line 89: consider deleting "which we assumed to be an underlying cause of the environmental degradation"

(b) Survey design

Move here material concerning the survey that is placed in other sections.

Line 98: consider replacing "mainly driven" with "facilitated by/through the collaboration with..."

Figure 1 caption: consider deleting "where surveys were carried out".

Line 100: Here "Xhosa" appears for the first time. Add a short explanation on how it is related to the case study area.

(c) Questionnaire design

Why are there separate subsections for Survey design and Questionnaire design?

Line 126: two tests

Lines 160-165: please check the consistency of the names of the seven/six demographic variables with their names in Table 1 and throughout the manuscript. Also, consider rephrasing in this direction: Seven demographic variables were recorded, and six quantified. Language...

Lines 174-177: Limitations of the used methods are mentioned.

General comment Methods: Try to make clearer how the three sections of your questionnaire (lines 109-112) answer your two research questions and/or aim.

Data collection and data analysis are satisfactorily distinguished.

I am not an expert in statistics, but the Methods section might benefit from more structure and clarity around which methods and tests were chosen to demonstrate what? E.g.: "In order to... , we applied... in relation to the first research question."

RESULTS

Line 190: consider rephrasing "to someone other than themselves". I remark the point made here is nicely discussed in the Discussion section (lines 271-281).

Lines 193-194: consider keeping this for discussion only: "which is indicative of potential gender-based power imbalances in the region".

Table 2 caption: consider separating in more sentences. Did the authors explain in the manuscript what is meant by the F-score?

General comment Results: Try to keep the same structure around the research questions, the threefold structure of the questionnaire (example: results to the analysis of section A of the questionnaire covering respondents' demographic information actually answer the second research question, etc.).

DISCUSSION

Lines 231-232: Consider rephrasing and check grammar: "Community tend not to self-organise when natural resources are very abundant or highly depleted [9,28]."

Line 238: Consider rephrasing and check grammar: "members will not follow rules today when future outcomes are uncertain".

Line 242 suggestion: "... " but also of the relationships between different members of the community and of their nature connectedness."

Line 284: replace "doomed to fail"

Line 323: suggestion for rephrasing: "they tend to attribute lower priority to these issues compared to technical solutions"

Line 325: Consider rephrasing: "Specifically, building on our results we make three recommendations on how to manage the... ". These are recommendations, or a general roadmap, as the subtitle suggests, rather than a fully laid out strategy.

General comment Discussion: The discussion is generally well written, and its development is consistent with the rest of the manuscript, including the abstract: "Therefore, we describe how strong leadership could nurture a sustainable social-ecological system by integrating ecological information and social empowerment into a multi-level governance system."

Decision letter (RSOS-180869.R0)

11-Oct-2018

Dear Dr Buschke:

Manuscript ID RSOS-180869 entitled "Ecosystem services and ecological degradation of communal wetlands in a South African biodiversity hotspot" which you submitted to Royal

Society Open Science, has been reviewed. The comments from reviewer(s) are included at the bottom of this letter.

In view of the criticisms of the reviewer(s), I must decline the manuscript for publication in Royal Society Open Science at this time. However, a new manuscript may be submitted which takes into consideration these comments.

Please note that resubmitting your manuscript does not guarantee eventual acceptance, and that your resubmission will be subject to re-review by the reviewer(s) before a decision is rendered.

You will be unable to make your revisions on the originally submitted version of your manuscript. Instead, revise your manuscript using a word processing program and save it on your computer.

You may also click the below link to start the resubmission process (or continue the process if you have already started your resubmission) for your manuscript. If you use the below link you will not be required to login to ScholarOne Manuscripts.

*** PLEASE NOTE: This is a two-step process. After clicking on the link, you will be directed to a webpage to confirm. ***

https://mc.manuscriptcentral.com/rsos?URL_MASK=cb20af2adbde4ad597e27f2579f48d35

Because we are trying to facilitate timely publication of manuscripts submitted to Royal Society Open Science, your resubmitted manuscript should be submitted by 10-Apr-2019. If you are unable to submit by this date please contact the Editorial Office for options.

Please note that Royal Society Open Science will introduce article processing charges for all new submissions received from 1 January 2018. Charges will also apply to papers transferred to Royal Society Open Science from other Royal Society Publishing journals, as well as papers submitted as part of our collaboration with the Royal Society of Chemistry (<http://rsos.royalsocietypublishing.org/chemistry>). If your manuscript is submitted and accepted for publication after 1 Jan 2018, you will be asked to pay the article processing charge, unless you request a waiver and this is approved by Royal Society Publishing. You can find out more about the charges at <http://rsos.royalsocietypublishing.org/page/charges>. Should you have any queries, please contact openscience@royalsociety.org.

I look forward to a resubmission.

on behalf of Prof. Kevin Padian (Subject Editor)
openscience@royalsociety.org

Associate Editor Comments to Author:

While recognising you have worked hard to address the concerns of the first round of review, the referees continue to have issues with your work, and do not feel able to recommend it for publication. As the journal's policy does not generally allow for multiple rounds of revision, we regret that we cannot consider the paper further for publication. Nevertheless, many thanks for considering Royal Society Open Science for your work, and we hope the referees' comments are valuable if you choose to submit the paper elsewhere.

Reviewer comments to Author:

Reviewer: 4

Comments to the Author(s)

The manuscript studies human-nature interactions using as a case study the ecological systems of wetlands and a rural community living in a biodiversity hotspot in South Africa as a social system. Interactions are based on benefits derived from the wetlands (ecosystem services) and damage-causing activities affecting the wetlands (ecological degradation). It brings to light: i) the relationship or lack of relationship between damage-causing activities, and benefits, and ii) the demographical factors that influence above interactions. The insights gained through this study are valuable as they cover an understudied, yet vulnerable area. However, the paper would benefit from a restructuring of content around the two enounced research questions. The manuscript went through the first round of revisions, and I acknowledge a lot of work has been invested in the statistical analysis, both from reviewers and authors. A general check for the English language is also recommended. Below are some specific comments.

ABSTRACT

The abstract correctly reflects the content of the paper.

Consider clarifying or separating the following statements for the abstract: "Respondents were more likely to blame others for wetland degradation, although there was no link between the damage-causing activities and benefits from wetlands." Is this reflected as such in the Results section?

INTRODUCTION

Lines 6-7: it is not clear how the stated goal "This study, therefore, set out to examine whether the people responsible for degrading wetlands are the same set of people who would benefit the most from their conservation" derives from the previous introductory sentences. Maybe delete "therefore" or make the link between motivation and aim clearer. Also, it is less common that the aim is stated so soon in a flow of a manuscript. Consider moving information about the aim of this study in the last paragraph of the introduction.

Lines 30-37: parts of this paragraph rather belong to section 2. Methods (a) Study area

Lines 38-65: most parts of these paragraphs rather belong to section 2. Methods (b) and (c)

Line 71: I am not sure "social capital" is a demographic characteristic. Consider adding "whether demographic characteristics and related social factors such as social capital, ..."

General comment Introduction: Consider restructuring the introduction so that each paragraph conveys a (general) message, that only foresees elements of the Discussion without revealing the results. For example, as a suggestion: a paragraph on wetlands in general, a paragraph on benefits/ecosystem services from wetlands, a paragraph on threats to wetlands in a certain context, (research gap), a paragraph on the aim and research questions.

METHODS

(a) Study area

Move here related material from the introduction.

Line 89: consider deleting “which we assumed to be an underlying cause of the environmental degradation”

(b) Survey design

Move here material concerning the survey that is placed in other sections.

Line 98: consider replacing “mainly driven” with “facilitated by/through the collaboration with...”

Figure 1 caption: consider deleting “where surveys were carried out”.

Line 100: Here “Xhosa” appears for the first time. Add a short explanation on how it is related to the case study area.

(c) Questionnaire design

Why are there separate subsections for Survey design and Questionnaire design?

Line 126: two tests

Lines 160-165: please check the consistency of the names of the seven/six demographic variables with their names in Table 1 and throughout the manuscript. Also, consider rephrasing in this direction: Seven demographic variables were recorded, and six quantified. Language...

Lines 174-177: Limitations of the used methods are mentioned.

General comment Methods: Try to make clearer how the three sections of your questionnaire (lines 109-112) answer your two research questions and/or aim.

Data collection and data analysis are satisfactorily distinguished.

I am not an expert in statistics, but the Methods section might benefit from more structure and clarity around which methods and tests were chosen to demonstrate what? E.g.: “In order to... , we applied... in relation to the first research question.”

RESULTS

Line 190: consider rephrasing “to someone other than themselves”. I remark the point made here is nicely discussed in the Discussion section (lines 271-281).

Lines 193-194: consider keeping this for discussion only: “which is indicative of potential gender-based power imbalances in the region”.

Table 2 caption: consider separating in more sentences. Did the authors explain in the manuscript what is meant by the F-score?

General comment Results: Try to keep the same structure around the research questions, the threefold structure of the questionnaire (example: results to the analysis of section A of the questionnaire covering respondents’ demographic information actually answer the second research question, etc.).

DISCUSSION

Lines 231-232: Consider rephrasing and check grammar: “Community tend not to self-organise when natural resources are very abundant or highly depleted [9,28].”

Line 238: Consider rephrasing and check grammar: “members will not follow rules today when future outcomes are uncertain”.

Line 242 suggestion: “...” but also of the relationships between different members of the community and of their nature connectedness.”

Line 284: replace “doomed to fail”

Line 323: suggestion for rephrasing: “they tend to attribute lower priority to these issues compared to technical solutions”

Line 325: Consider rephrasing: “Specifically, building on our results we make three recommendations on how to manage the...”. These are recommendations, or a general roadmap, as the subtitle suggests, rather than a fully laid out strategy.

General comment Discussion: The discussion is generally well written, and its development is consistent with the rest of the manuscript, including the abstract: “Therefore, we describe how strong leadership could nurture a sustainable social-ecological system by integrating ecological information and social empowerment into a multi-level governance system.”

Reviewer: 2

Comments to the Author(s)

The specific comments to the manuscript are:

This manuscript is about the ecosystem services and ecological degradation of the communal wetlands, but nowhere in the manuscript the authors have provided the specific ecosystem services to the local community of the study area. The authors have framed this manuscript around the perceived benefits to self as well as other. But how did the authors choose just a few of the provisioning (plant and water), regulating (flood and erosion) and cultural services, as mentioned in the questionnaire design.

What was the reason that most of the respondents were in their 20s and 30s, and how many were leaders (traditional or formal), as in survey design it is mentioned that community survey was conducted with community members and their leadership.

Is community leader a form of employment? I don't agree.

To know demographic composition, it is advised to refer to the census report for the area, which will give accurate information on the demographic structure as well as economic and employment status.

It will be useful to provide a list of traditional activities carried out by people in the wetlands, in the Study Area section.

Page 4; lines 47-48: "According to the IDP [22], the local community has low levels of education, which we assumed 89 to be an underlying cause of the environmental degradation."

Low level of education doesn't necessarily always results in environment degradation. Moreover, education doesn't grant the person to be environmentally conscious.

Page 6; Line 48: ".....All forms of bene?ts were accrued more than occasionally....", delete 'were'

Page 6; lines 53-54: "....The multiple regression model showed that male respondents were more likely to bene?t from wetland ecosystem services (Table 2),"

Table 2 shows damage causing activities, whereas the sentence says that male respondents were more likely to benefit from wetland ecosystem.

The language is lax and needs proper editing for grammatical and syntax errors.

Author's Response to Decision Letter for (RSOS-180869.R0)

See Appendix B.

RSOS-181770.R0

Review form: Reviewer 5

Is the manuscript scientifically sound in its present form?

Yes

Are the interpretations and conclusions justified by the results?

Yes

Is the language acceptable?

Yes

Is it clear how to access all supporting data?

Yes

Do you have any ethical concerns with this paper?

No

Have you any concerns about statistical analyses in this paper?

I do not feel qualified to assess the statistics

Recommendation?

Accept with minor revision (please list in comments)

Comments to the Author(s)

GENERAL COMMENTS

I must preface my review by noting that I am not experienced in questionnaire surveys and their analysis and interpretation, and therefore I am very limited in terms of commenting on any methodological issues in the paper. Instead I comment mainly from the perspective of my experience in assessing wetland ecological degradation and ecosystem services provision.

The manuscript addresses a relevant and important topic for which there has not been much research conducted in South Africa. From my perspective (and with my limitations in terms of questionnaire surveys) the study appears to be generally scientifically sound, well conceptualized and implemented, and well presented, and makes a useful scientific contribution with practical application. However, there are three important general issues which should be addressed, as well a few specific issues which require attention, which are given below.

Elaboration on damage to wetlands and the basis on which specific activities are deemed to be damaging or not:

Although the paper identifies four damage-causing activities fairly early on in the paper (Grazing, Ploughing, Dumping and Burning) it lacks any specific indication of what is meant by damage to wetlands and the basis on which specific activities are deemed to be damaging or not. It is important that this be done given that damage-causing activities are a central focus of the paper. The implication seems to be that all of the four listed uses are inherently damaging, and while this probably stands for ploughing and dumping, it is more nuanced for grazing and burning. Wetlands over much of South Africa, and probably in the study area, evolved under fire and indigenous grazers (e.g. buffalo) (Fynn et al. 2015 and Kotze 2013) and where large grazers such as buffalo are no longer present in an area then domestic livestock can to fair degree

replace their effect. The damage resulting from fire and grazing relates to the specifics of the grazing/burning regime, e.g. from annual fires or prolonged intense grazing.

Fynn RWS, Murray-Hudson M, Dhliwayo M, Scholte P, 2015. African wetlands and their seasonal use by wild and domestic herbivores. *Wetlands, Ecology and Management* 23:559–581

Kotze D C, 2013. The effects of fire on wetland structure and functioning. *African Journal of Aquatic Science* 38: 237–247

A greater depth of reporting on the different wetland uses/benefits:

A greater depth of reporting is required on the different uses made of the wetlands in the study area, the compatibility/incompatibility of these respective uses and how demographics (in particular gender) may be specific to particular uses. This would help given the reader more insights into some of the key findings reported in the paper, e.g. that males were more likely to benefit from the wetland than females.

Also see specific comments on Page 7, Figure 3.

Reporting on collective choice rules for wetlands in the study area:

The paper makes several general references to collective choice rules and has a sub-section in the discussion on Governance. One would therefore anticipate that some specifics on these rules would be reported on for wetlands in the study area. But this seems to be entirely missing. While acknowledging that this was not the focus of the questionnaire, it is relevant to the study objectives and deserves some attention, even if it is that no collective choice rules for wetlands in the study area appear to exist.

SPECIFIC COMMENTS

Abstract, “However, wetlands are often on communal land and face degradation by individuals who maximise their personal benefit from ecosystem services without bearing the full environmental costs of their actions.”

I suggest deleting this sentence, which is general and I am also not sure fully substantiated. This would provide more words available in the Abstract for the authors to report on an additional specific finding of the research in place of this general statement.

Page 2, line 15-17, “Common-pool resources are resources that are rival (i.e. 16 using the resources reduces its availability to other users), but non-excludable (i.e. one cannot 17 restrict access to these resources)”

It is not necessarily so that common-pool resources are non-excludable, as in fact is identified in the following paragraph’s reference to collective choice rules (which may include exclusions).

Page 3, line 72, “These wetlands are....prone to degradation caused by man-made activities.” /A few words indicating why they are prone to such degradation would be useful

Page 4 line 100-104. “(2) how often respondents partook in the most common damage-causing activities to 1 wetlands (identified during prior site visits)

A few words should be added in terms of what was carried out in the “prior site visits”. As it stands, this is not clear.

Page 6, line 189, “Notably, respondents who were more likely to partake in grazing, ploughing, dumping or burning did not stand to benefit any more from wetland ecosystem services than their peers.” Sorry this is a little unclear. Are the “peers” being referred to here not likely to partake in grazing, ploughing, dumping or burning? This should be clarified.

Page 7, para 1: “perceive group” should read “perceived group”

Page 7, Figure 3: In the text accompanying Figure 3 it would be informative to report on some of the specific services within the three broad categories reported in /Figure 3, namely provisioning, regulating and cultural services. As it stands I cannot see it reported anywhere in the paper.

Page 8, line 193, “This study showed that all community members benefited from ecosystem services from wetlands in the Hlabathi administrative area, South Africa,” This statement could be better substantiated through more specific reporting.

Page 8, line 193-195, “This study showed that all community members benefited from ecosystem services from wetlands in the Hlabathi administrative area, South Africa, but men tended to benefit more than women.” This statement needs to be substantiated and elaborated on with more specific reporting, in particular giving the reader a better sense of which of the specific services this applied to, thereby helping the reader to better understand why men tended to benefit more than women.

Page 8, line 197-199, “Moreover, there was no evidence that the men and women who degraded wetlands gained more from wetland ecosystem services than their fellow community members.” More information is required in terms of what constitutes someone who degraded wetlands. See General comments.

Page 8, line 216: I am not sure that “seasonal grazing” is a good example of a resource which is mobile, which seems to be implied.

Page 9 line 218-219, “However, the productivity and predictability of these ecosystems might be potential obstacles to their sustainable use.” This statement needs to be better elaborated upon, e.g. productivity and predictability in terms of what?

Page 9 line 242-244, “Community members have a shared interest in preserving wetland systems, which frees them from the conflict caused by different land-use pressures (e.g. [35,36]).” Sorry, it is unclear to me while a general shared interest in preserving wetlands should free community members from conflict caused by different land-use pressures. I can appreciate how a shared interest could help in resolving conflicts, but it seems unrealistic that it would free users from conflicts.

Page 9 line 248, “Gender is an important determinant of pro-environmental behaviour” Some elaboration of why this should be so would be useful. In addition, what would constitute pro-environmental behaviour?

Page 10 line 279, “The final predictor of sustainable use of common-pool resources is good governance structures with collective choice rules” As elaborated on in the General comments, the paper provides important general statements such as this but seems to be lacking in terms of reporting on the specifics of any collective choice rules for use of the wetlands in the study area.

Page 10 line 286, “This is not because respondents are generally selfish or have negative perceptions of nature. On the contrary, respondents were Xhosa, a cultural group that has been shown previously to score highly on various measures of nature connectedness [27,39]. Such human-biosphere connectedness is central to sustainable use of ecosystems and resilient social-ecological systems [48,49].” Some of the assumptions in these statements are potentially problematic, in particular what appears to be an assumed overriding influence that so-called “nature connectedness” might have over immediate self interests.

Page 10 line 292, “It is possible that communicating scientific information to traditional leadership structures could establish sustainable practices throughout the wider community.” What appears to be the implicit assumption here may often not hold that altered awareness/perceptions will necessarily lead to altered practices.

Page 10 line 299 “Instead, it needs to be embedded into a multi-level governance structure (e.g. [52]) that includes national initiatives, like the Working for Wetlands expanded public works programme” I am not sure that a public works programme is the best example to give for a natural resources governance structure.

Page 12 line 389 “Institute SANB. 2016 Classification system for wetlands and other aquatic ecosystems. Pretoria, South Africa: Government Printer.”

I suspect that this reference has been incorrectly cited and should appear as follows: “Ollis DJ, Snaddon CD, Job NM, Mbona N 2013. Classification System for Wetlands and other Aquatic Ecosystems in South Africa. User Manual: Inland Systems No. 22, SANBI Biodiversity Series. South African National Biodiversity Institute, Pretoria. Draft final report to the Water Research Commission, Pretoria.”

Decision letter (RSOS-181770.R0)

22-Feb-2019

Dear Dr Buschke

On behalf of the Editor, I am pleased to inform you that your Manuscript RSOS-181770 entitled "Ecosystem services and ecological degradation of communal wetlands in a South African biodiversity hotspot" has been accepted for publication in Royal Society Open Science subject to minor revision in accordance with the referee suggestions. Please find the referees' comments at the end of this email.

The reviewers and Subject Editor have recommended publication, but also suggest some minor revisions to your manuscript. Therefore, I invite you to respond to the comments and revise your manuscript.

- Ethics statement

- Data accessibility

It is a condition of publication that all supporting data are made available either as supplementary information or preferably in a suitable permanent repository. The data accessibility section should state where the article's supporting data can be accessed. This section should also include details, where possible of where to access other relevant research materials such as statistical tools, protocols, software etc can be accessed. If the data has been deposited in an external repository this section should list the database, accession number and link to the DOI for all data from the article that has been made publicly available. Data sets that have been

deposited in an external repository and have a DOI should also be appropriately cited in the manuscript and included in the reference list.

If you wish to submit your supporting data or code to Dryad (<http://datadryad.org/>), or modify your current submission to dryad, please use the following link:
<http://datadryad.org/submit?journalID=RSOS&manu=RSOS-181770>

- **Competing interests**

- **Authors' contributions**

- **Acknowledgements**

- **Funding statement**

Because the schedule for publication is very tight, it is a condition of publication that you submit the revised version of your manuscript before 03-Mar-2019. Please note that the revision deadline will expire at 00.00am on this date. If you do not think you will be able to meet this date please let me know immediately.

When submitting your revised manuscript, you will be able to respond to the comments made by the referees and upload a file "Response to Referees" in "Section 6 - File Upload". You can use this

to document any changes you make to the original manuscript. In order to expedite the processing of the revised manuscript, please be as specific as possible in your response to the referees.

on behalf of Professor Kevin Padian (Subject Editor)
openscience@royalsociety.org

Associate Editor Comments to Author:

Please accept our apologies for the delay in completing review of your manuscript: the original reviewers were unable to assist, and it has taken some time to secure the advice of the reviewer here (though we're grateful for all the support of the reviewers).

The comments of the referee are broadly positive but the Editors want to emphasise two points in particular to the authors:

1. Regarding your dataset/code: you indicate you will make the material available via FigShare on acceptance. As the Royal Society Publishing provides this service free for authors' electronic supplements, please ensure these are included in your revision. This is an absolute requirement of publication;

2. The Editors do not consider the referee's comments to preclude publication, but you **MUST** respond to their comments. You should include the changes and modifications requested, and also provide full responses in your reply to reviewers.
Good luck, and we'll await your revision.

Reviewer comments to Author:

Reviewer: 5

Comments to the Author(s)

GENERAL COMMENTS

I must preface my review by noting that I am not experienced in questionnaire surveys and their analysis and interpretation, and therefore I am very limited in terms of commenting on any methodological issues in the paper. Instead I comment mainly from the perspective of my experience in assessing wetland ecological degradation and ecosystem services provision.

The manuscript addresses a relevant and important topic for which there has not been much research conducted in South Africa. From my perspective (and with my limitations in terms of questionnaire surveys) the study appears to be generally scientifically sound, well conceptualized and implemented, and well presented, and makes a useful scientific contribution with practical application. However, there are three important general issues which should be addressed, as well a few specific issues which require attention, which are given below.

Elaboration on damage to wetlands and the basis on which specific activities are deemed to be damaging or not:

Although the paper identifies four damage-causing activities fairly early on in the paper (Grazing, Ploughing, Dumping and Burning) it lacks any specific indication of what is meant by damage to wetlands and the basis on which specific activities are deemed to be damaging or not. It is important that this be done given that damage-causing activities are a central focus of the paper. The implication seems to be that all of the four listed uses are inherently damaging, and while this probably stands for ploughing and dumping, it is more nuanced for grazing and burning. Wetlands over much of South Africa, and probably in the study area, evolved under fire and indigenous grazers (e.g. buffalo) (Fynn et al. 2015 and Kotze 2013) and where large grazers such as buffalo are no longer present in an area then domestic livestock can to fair degree replace their effect. The damage resulting from fire and grazing relates to the specifics of the grazing/burning regime, e.g. from annual fires or prolonged intense grazing.

Fynn RWS, Murray-Hudson M, Dhliwayo M, Scholte P, 2015. African wetlands and their seasonal use by wild and domestic herbivores. *Wetlands, Ecology and Management* 23:559–581

Kotze D C, 2013. The effects of fire on wetland structure and functioning. *African Journal of Aquatic Science* 38: 237–247

A greater depth of reporting on the different wetland uses/benefits:

A greater depth of reporting is required on the different uses made of the wetlands in the study area, the compatibility/incompatibility of these respective uses and how demographics (in particular gender) may be specific to particular uses. This would help given the reader more insights into some of the key findings reported in the paper, e.g. that males were more likely to benefit from the wetland than females.

Also see specific comments on Page 7, Figure 3.

Reporting on collective choice rules for wetlands in the study area:

The paper makes several general references to collective choice rules and has a sub-section in the discussion on Governance. One would therefore anticipate that some specifics on these rules would be reported on for wetlands in the study area. But this seems to be entirely missing. While acknowledging that this was not the focus of the questionnaire, it is relevant to the study objectives and deserves some attention, even if it is that no collective choice rules for wetlands in the study area appear to exist.

SPECIFIC COMMENTS

Abstract, "However, wetlands are often on communal land and face degradation by individuals who maximise their personal benefit from ecosystem services without bearing the full environmental costs of their actions."

I suggest deleting this sentence, which is general and I am also not sure fully substantiated. This would provide more words available in the Abstract for the authors to report on an additional specific finding of the research in place of this general statement.

Page 2, line 15-17, "Common-pool resources are resources that are rival (i.e. 16 using the resources reduces its availability to other users), but non-excludable (i.e. one cannot 17 restrict access to these resources)"

It is not necessarily so that common-pool resources are non-excludable, as in fact is identified in the following paragraph's reference to collective choice rules (which may include exclusions).

Page 3, line 72, "These wetlands are....prone to degradation caused by man-made activities." / A few words indicating why they are prone to such degradation would be useful

Page 4 line 100-104. "(2) how often respondents partook in the most common damage-causing activities to1 wetlands (identified during prior site visits)

A few words should be added in terms of what was carried out in the "prior site visits". As it stands, this is not clear.

Page 6, line 189, "Notably, respondents who were more likely to partake in grazing, ploughing, dumping or burning did not stand to benefit any more from wetland ecosystem services than their peers." Sorry this is a little unclear. Are the "peers" being referred to here not likely to partake in grazing, ploughing, dumping or burning? This should be clarified.

Page 7, para 1: "perceive group" should read "perceived group"

Page 7, Figure 3: In the text accompanying Figure 3 it would be informative to report on some of the specific services within the three broad categories reported in /Figure 3, namely provisioning, regulating and cultural services. As it stands I cannot see it reported anywhere in the paper.

Page 8, line 193, "This study showed that all community members benefited from ecosystem services from wetlands in the Hlabathi administrative area, South Africa," This statement could be better substantiated through more specific reporting.

Page 8, line 193-195, "This study showed that all community members benefited from ecosystem

services from wetlands in the Hlabathi administrative area, South Africa, but men tended to benefit more than women.” This statement needs to be substantiated and elaborated on with more specific reporting, in particular giving the reader a better sense of which of the specific services this applied to, thereby helping the reader to better understand why men tended to benefit more than women.

Page 8, line 197-199, “Moreover, there was no evidence that the men and women who degraded wetlands gained more from wetland ecosystem services than their fellow community members.” More information is required in terms of what constitutes someone who degraded wetlands. See General comments.

Page 8, line 216: I am not sure that “seasonal grazing” is a good example of a resource which is mobile, which seems to be implied.

Page 9 line 218-219, “However, the productivity and predictability of these ecosystems might be potential obstacles to their sustainable use.” This statement needs to be better elaborated upon, e.g. productivity and predictability in terms of what?

Page 9 line 242-244, “Community members have a shared interest in preserving wetland systems, which frees them from the conflict caused by different land-use pressures (e.g. [35,36]).” Sorry, it is unclear to me while a general shared interest in preserving wetlands should free community members from conflict caused by different land-use pressures. I can appreciate how a shared interest could help in resolving conflicts, but it seems unrealistic that it would free users from conflicts.

Page 9 line 248, “Gender is an important determinant of pro-environmental behaviour” Some elaboration of why this should be so would be useful. In addition, what would constitute pro-environmental behaviour?

Page 10 line 279, “The final predictor of sustainable use of common-pool resources is good governance structures with collective choice rules” As elaborated on in the General comments, the paper provides important general statements such as this but seems to be lacking in terms of reporting on the specifics of any collective choice rules for use of the wetlands in the study area.

Page 10 line 286, “This is not because respondents are generally selfish or have negative perceptions of nature. On the contrary, respondents were Xhosa, a cultural group that has been shown previously to score highly on various measures of nature connectedness [27,39]. Such human-biosphere connectedness is central to sustainable use of ecosystems and resilient social-ecological systems [48,49].” Some of the assumptions in these statements are potentially problematic, in particular what appears to be an assumed overriding influence that so-called “nature connectedness” might have over immediate self interests.

Page 10 line 292, “It is possible that communicating scientific information to traditional leadership structures could establish sustainable practices throughout the wider community.” What appears to be the implicit assumption here may often not hold that altered awareness/perceptions will necessarily lead to altered practices.

Page 10 line 299 “Instead, it needs to be embedded into a multi-level governance structure (e.g. [52]) that includes national initiatives, like the Working for Wetlands expanded public works programme” I am not sure that a public works programme is the best example to give for a natural resources governance structure.

Page 12 line 389 “Institute SANB. 2016 Classification system for wetlands and other aquatic ecosystems. Pretoria, South Africa: Government Printer.”

I suspect that this reference has been incorrectly cited and should appear as follows: “Ollis DJ, Snaddon CD, Job NM, Mbona N 2013. Classification System for Wetlands and other Aquatic Ecosystems in South Africa. User Manual: Inland Systems No. 22, SANBI Biodiversity Series. South African National Biodiversity Institute, Pretoria. Draft final report to the Water Research Commission, Pretoria.”

Author's Response to Decision Letter for (RSOS-181770.R0)

See Appendix C.

RSOS-181770.R1 (Revision)

Review form: Reviewer 5

Is the manuscript scientifically sound in its present form?

Yes

Are the interpretations and conclusions justified by the results?

Yes

Is the language acceptable?

Yes

Is it clear how to access all supporting data?

Yes

Do you have any ethical concerns with this paper?

No

Have you any concerns about statistical analyses in this paper?

I do not feel qualified to assess the statistics

Recommendation?

Accept as is

Comments to the Author(s)

As far as I can see the authors have done well in addressing all of the issues I raised in the previous draft.

Decision letter (RSOS-181770.R1)

14-May-2019

Dear Dr Buschke,

I am pleased to inform you that your manuscript entitled "Ecosystem services and ecological degradation of communal wetlands in a South African biodiversity hotspot" is now accepted for publication in Royal Society Open Science.

on behalf of Prof Kevin Padian (Subject Editor)
openscience@royalsociety.org

Reviewer comments to Author:
Reviewer: 5

Comments to the Author(s)
As far as I can see the authors have done well in addressing all of the issues I raised in the previous draft.

Appendix A

Dear editors,

Thank you for the opportunity to resubmit this manuscript to Royal Society Open Science. As you will see from our detailed responses below, we have taken the valuable feedback from the reviewers and incorporated it into this revised manuscript. We are confident that we have addressed their concerns and that this resubmission is an improvement on our original submission.

We would like to thank the reviewers, Mark Difford and Ruchi Badola, of their thorough advice, which has made a substantial contribution to this new manuscript.

We hope that you find our amendments adequate and we keenly await your decision.

Kind regards,
Falko Buschke

What follows is a point-by-point reply to the reviewers comments.

#####

Subject Editor Comments to Author:

One reviewer was positive but offered no substantial comments. The other two reviewers found severe problems which in their views would prevent publication of the manuscript. We will allow a resubmission, if the authors choose, but only if all the criticisms of the referees are adequately assessed, which may take considerable additional work. Thanks for submitting here and good luck.

#####

Reply 1: Thank you and the three reviewers for the constructive feedback on this manuscript. We have carried out considerable revisions based on the feedback, which we highlight in yellow for ease of reference. In instances where we followed the reviewers suggestions fully, we list the line number of these changes in our replies to each of their remarks. When we did not incorporate their feedback, we explain in this letter why their advice is not reflected in our revised manuscript.

Briefly, we have implemented the analyses suggested by Reviewer 1, which changed our conclusions somewhat. The new analyses demonstrate that the only significant explanatory variable for benefits from wetland ecosystem services is the gender of the respondent (males report more benefits than females). This change has meant that the abstract and discussion sections had to be modified. Besides this major change, the other alternations to the manuscript were intended to add clarity, context or additional information to the original submission, which improves the readability, but does not change the main take-home messages.

#####

Reviewers' Comments to Author:

Reviewer: 1

Comments to the Author(s)

Overview: ---

This is a relatively well written paper that presents the results of a study of wetland-use by a community living in an ecologically sensitive part of South Africa. The dataset on which the results are based is valuable, and the results of the study are likely to be of some interest even to a general audience. Publication of the paper in its present form cannot however be recommended because (1) inappropriate methods have been used for some of the analyses; (2) the results of some of the analyses are incorrectly interpreted; and (3) the scales that have been used to measure benefit from and damage to wetlands are insufficiently unidimensional, and to some extent are conflated (they partly measure each other). Once these problems have been fixed, and parts of the paper rethought, I think that the results would be worth publishing.

Below I briefly review the main problems I identified, beginning with a set of lesser problems. In the last part of my review, under the head Detailed Comments, I address these and other issues in greater detail and more directly, on a line-by-line basis. I have also attached a report in which I provide further proofs of most of the key issues and code to support the alternative analyses that I propose.

#####

Reply 2: We would like to thank Reviewer 1 for the incredibly detailed review; it was the most thorough

evaluation of research we have experienced so far. We have tried to incorporate as much of the feedback as possible, but we had to make some necessary trade-offs because, as we are sure the reviewer will acknowledge, statistical analyses are rarely perfect and require compromises on the underlying assumptions. Most notable, we still analysed the data using OLS regression, because we are of the view that the benefits of including multiple explanatory variable outweigh the negatives of treating survey-based responses as continuous variables.

#####

Relatively Minor, but Nonetheless Important Problems Are:---

1) The authors treat the individual items of each scale, i.e. the 16 questions of their survey, as if they are interval- or ratio-level variables, i.e. continuous variables. However, each question is an ordinal-level variable (an ordered factor) with just three categories (Never, Sometimes, Always). This affects their calculation of statistics on the reliability, validity, and dimensionality of each scale, where Pearson correlations are used. Polychoric correlations should have been used instead. Revelle's omega function, which the authors used to calculate these statistics, uses Pearson correlations by default, but there is an argument to that function that allows polychoric correlations to be used (in the event that scale items have fewer than ~ seven categories or levels). Incorrectly treating individual items as if they are continuous variables also affects the results reported in Table 3, where the means and standard errors of individual questions are calculated and reported. In my Detailed Comments I propose a replacement analysis, which I flesh out in my attached report. Good options are to use Wilcoxon signed-rank tests or Cliff's delta statistics (R package orddom). See Carifio & Perla (2007) and Brown (2011) for notes on the distinction between the items of a scale and the scale itself, and what the acceptable numerical treatment of each is. Also see Cliff (1993, 1996).

#####

Reply 3: We have followed this suggestion and have changed our analyses to non-parametric equivalents. As suggested, we now used a Wilcoxon test to compare self-reported and perceived group benefits from and damage to wetlands (lines 134-137). Furthermore, to compare different types of ecosystem services (Figure 3), we use a Friedman test, with appropriate post-hoc tests, which is the non-parametric equivalent of a repeated measures anova (explained in lines 120-127).

With reference to our original Table 3 (an overview of the different types of damage-causing activities): this has been removed from our revised manuscript because it contained information non-essential to our main discussion. Thus, the problems pointed out by the reviewer does not apply to this revised manuscript.

#####

2) The authors fit no-intercept regressions (but do not tell the reader); the results are reported in Table 2. They then make the mistake of interpreting the statistics that result from those regressions as if they derive from regressions that were fitted with an intercept term. R^2 (coefficient of determination) values from intercept- and no-intercept regressions have fundamentally different interpretations (true for all reputable statistical software programs). Notably, the R^2 value from a no-intercept regression does not represent the amount of variance of the regressand (response) that the regressor (predictor) explains; it also does not represent the extent to which the regressand and the regressor are associated. This is relatively easy to demonstrate. The Pearson correlation coefficient (measures linear association) between the regressor and the regressand of the first regression listed in Table 2 (Benefits from wetlands) is 0.140. If we square this we get the R^2 for an ordinary least-squares model fitted with an intercept term, which is 0.020, compared to 0.944 for the no-intercept regression. Given that the accepted interpretation of models fitted with an intercept term represents the proportion of variance in the response variable that the predictor explains (here ~ 2% compared to ~ 94% for the no-intercept regression), R^2 values from no-intercept models cannot, ipso facto, have the same interpretation but must have a wholly different one. The R^2 value from a no-intercept regression in fact measures the proportion of the variability _about the origin_ that the regression explains, and not the proportion of variability of the response variable as a whole, or around the grand mean, that the regression explains. See Hocking (1996) and Eisenhauer (2003) and elsewhere.

Additionally, the estimated slope-coefficients of no-intercept models depend on the strong theoretical assumption that the expected value of the response variable is zero when the value of the predictor variable is zero. The authors provide no evidence to support such a claim, which, indeed, seems to be part of what they are trying prove. For the first regression reported in Table 2, the slope coefficient for an intercept-based regression is 0.10 not 0.88. It might also be noted that the estimates reported for the first regression (Benefits from wetlands) are in any case wrong because the authors mistakenly

included two questions, viz Q5 and Q11, in both the regressor and the regressand when calculating the mean-scores of sub-scales that they use in the regression.

While it might be of some interest to know how the two subscales of each scale relate to each other (i.e. how the self-assessed and perceived-group effects relate to each other), this type of analysis should not be necessary if the main scales are sharpened (or refined) to be truly unidimensional. The authors would probably be better off carrying out some form of agreement/concordance analysis, though Lin's concordance correlation coefficient (Lin 1989; Lin et al. 2002) is probably too exacting. Bland-Altman (i.e. Tukey's mean-difference) plots, coupled with paired t-tests or a non-parametric or Bayesian alternative, would probably be adequate. See the R packages granovaGG (function granovagg.ds), epiTools, MethComp, Agreement, amongst others.

#####

Reply 4: We agree that our original analysis was crude and have implemented his proposed improvements. We used dependent sample assessment plots (new Figure 2 using the 'granovagg.ds' command as suggested: line 137-138) in combination with Wilcoxon non-parametric tests (lines 134-137) as suggested to compare self- and group-reported benefits and damages to wetlands. We are pleased that this improved analysis supported our original conclusions – that respondents underestimate their own damage to wetlands, but accurately assess the benefits they gain – in a way that will be clearer to the readers. This new analysis enhances this revised manuscript.

We have also corrected the mistake of including two questions on both the regressor and regressed scales (see R-scripts on Github). Furthermore, we refined the two scales and ensured that they are indeed unidimensional as outlined in lines 148-160 (we elaborate on this process in a reply below).

#####

This brings me to a final issue under this head. Ordinary least-squares regression is not appropriate when both the regressor and the regressand are measured with approximately the same degree of error, as they are here. The methods I have just referred to, viz agreement/concordance analysis, use major-axis (or Deming) regression as a central tool. It is the appropriate type of regression to use when regressing one psychometric or survey-based scale on another one. See R packages lmodel2 and smatr and references therein.

#####

Reply 5: We appreciate this comment by the referee, but, regrettably, we unable to implement the suggestion. To our understanding, major-axis regression (in packages lmodel2 and smatr) can only be used for univariate regression. Since it was important that we incorporated both continuous and categorical demographic explanatory variable in our model (Table 2), we could not use the more robust major-axis regression. Instead, we chose to stick with the OLS regression. While this is not ideal, as the reviewer points out, we feel it is an acceptable compromise. In our model outputs (Table 2), only one explanatory variable is statistically significant (Gender, $P = 0.005$) and its P-value is safely below the conventional threshold of $\alpha = 0.05$. Even though OLS regression might be biased, we argue that this bias is an acceptable compromise to ensure that we can model all demographic variables simultaneously. This is included in lines 175-178.

#####

3) The authors use pairwise correlations to assess the collinearity of putative predictors for the model they summarize in Figure 3 and Table 4. Pairwise correlations are neither an appropriate nor a recommended way of assessing collinearity, and do not provide grounds for excluding variables from the model. The reason of this is that collinearity depends on jointly assessing all the predictors in the model. Recommended approaches rely on (generalized) variance-inflation factors and condition indices of the design matrix (Belsley et al. 1980 [2004]). See vif() in Fox's car package for R, and ols_coll_diag() in Hebbali's olsrr package for R.

Were the authors to use these methods to test for collinearity they would find that neither of the two variables they exclude come close to showing signs of being collinear with other variables. In fact, the most collinear putative predictor is Position (role played in the community), which has a generalized variance-inflation factor of 2.64. This is well below the recommended value of >4 , which is when one begins to suspect that there might be a sufficient degree of collinearity to begin to affect parameter estimates. Values above 10 are generally recognized as indicating a degree of collinearity that needs fixing.

#####

Reply 6: We agree with the reviewer on this point that the variance inflation factors are not high enough to imply collinearity in predictor variables. We have, therefore, included all the demographic variable in our regression model (see lines 171-173), outputs from which are included in Table 2.

#####

The Main Problem:---

The authors results and conclusions rest on the reliability, unidimensionality, and validity of the two scales they use. Having explored these scales in considerable detail while preparing this review I am not convinced by the authors' argument that the scales are sufficiently unidimensional for mean-scores (that have a clear interpretation) to be formed, and therefore for the analysis to proceed.

In supporting their argument, the authors cite Wilhelm-Rechmann et al. (2014) [which I was a coauthor of] and Reise et al. (2010). However, there are fundamental differences between the arguments of Wilhelm-Rechmann et al. (2014) and the present study, notably: (1) the statistics on unidimensionality reported by Wilhelm-Rechmann et al. are better than those reported here; (2) the scale Wilhelm-Rechmann et al. used, the NEP (New Environmental Paradigm) scale, is a known quantity that has been used in hundreds of studies worldwide with reliable results, despite showing signs of being weakly multidimensional; (3) five-point items, which the NEP scale uses, are more likely to give the spurious impression of multidimensionality than three-point items are; and (4) Wilhelm-Rechmann et al. were able to demonstrate the robustness of their findings by favourably comparing them with (a) the results of a meta-analysis of 69 studies spanning 36 nations, and (b) the results of Dunlap et al.'s survey of residents of Washington State (USA). See Wilhelm-Rechmann et al. (2014) for details. The authors of the present paper unfortunately do not have access to such "back-stop" studies that might be used to sway the argument or corroborate their results.

In rejecting the authors' arguments on this fundamentally important question (how much multidimensionality is too much) I have additionally been swayed by the following facts. In deciding whether a multidimensional scale is sufficiently unidimensional for analysis to proceed safely, Reise (see the reference list and the help file for Revelle's omega function in R) stresses an index known as the explained common variance (ECV), which he calls a 'cleaner index of "unidimensionality"' than omega hierarchical (Reise et al. 2010). He and his associates recommend that this be > 0.70 , except if the percentage of uncontaminated correlations (PUC) is high (> 0.80), in which case an $ECV > 0.50$ would be acceptable (Reise et al. 2016). When PUC is less than 0.80 then Reise and associates recommend that $ECV > 0.60$ and that omega hierarchical be > 0.70 . Only then can one be reasonably sure that the presence of some multidimensionality is not severe enough to disqualify the interpretation of the scale as being primarily unidimensional (Reise et al. 2013).

The problem is that the authors' scales have ECV and PUC values of 0.38/0.36 (Pearson/polychoric correlations) and 0.64/0.55, respectively, for the benefits scale, and ECV and PUC values of 0.34/0.36 and 0.56/0.56, respectively, for the damage-focused scale. Revelle's beta coefficient for the two scales is also unacceptably low, being 0.26/0.20 and 0.25/0.07, respectively. At the very least, it should be > 0.5 , and preferably should be > 0.7 (Revelle 1979, Cooksey & Soutar 2006). In short, there is no evidence to support the authors' view that their scales are sufficiently unidimensional for them to be treated as if they are unidimensional, and hence for mean (or summated) scores to have a clear interpretation.

As they stand, the two scales are moderately correlated ($r = 0.378$). This indicates a degree of conflation between them, a suspicion that is confirmed by exploratory analysis using tools designed to select unidimensional scales. Mokken Scale Analysis (R package mokken), for instance, excludes Q5, Q11, and Q15 from any scale, and puts three items from the scale to assess damage to the wetland (Q2, Q8, Q16) on the scale to assess benefits from the wetland, and it puts one item from the scale to assess benefits from the wetland (Q9) on the scale to assess damage to the wetland. These Mokken-based scales, with mixed questions from the original two scales. have considerably better statistics on reliability and dimensionality than the scales formed by the authors; they also are not correlated ($r = 0.091$).

Using a set of tools that I routinely use, I was able to manually select items from each scale that are indisputably unidimensional, based on the criteria suggested by Reise and associates (op. cit.). Significantly most of the items that I excluded, viz Q5, Q7, Q9, Q2, Q8, and Q16, are items that the foregoing exploratory methods either excluded or put on the other scale, viz Q9 on one hand, and Q2,

Q8, and Q16 on the other. The two scales that are formed after these exclusions[*] are not correlated ($r = 0.072$) and have omega hierarchical, ECV, and PUC statistics (based on polychoric correlations) of 0.71, 0.68, and 0.67 for the benefit-scale, and 0.69, 0.63, and 0.53 for the damage-scale. These are good scale-statistics for data of this type.

[*] Revised benefits scale: [Q1,Q3,Q11,Q13,Q15] (excludes Q5,Q7,Q9); revised damage scale: [Q4,Q6,Q10,Q12,Q14] (excludes Q2,Q8,Q16).

The foregoing findings mean that were the authors prepared to revise their scales along the lines I have proposed, there is much that they could rescue from this interesting study (and worthwhile dataset) and be confident about the findings. There is no reason, I think, why at least some of the excluded items (i.e. questions), namely those that measure clearly defined activities like grazing (Q2) and burning (Q8, Q16), could not be used as ancillary variables in supporting analyses. For instance, they could perhaps be used as predictors in the replacement-analysis I suggest for Table 2. See the expository document/code that I have attached for ideas and supporting proofs.

#####

Reply 7: We would like to thank the reviewer for this suggestion and especially for the thorough supplementary notes he provided. We would not have been able to implement these suggestion without the step-by-step guidance in the form of annotated R-scripts. We are very grateful for this assistance.

There is little to reply other than we have implemented this suggestion exactly as the reviewer outlined. It is mentioned briefly in lines 148-160, but it was not possible for use to include as much elaborate discussion in the manuscript as provided here. Nevertheless, the new scale used in this revision are the same as the ones proposed by the referee, and we used the same metrics of unidimensionality (Cronbach's alpha, Hierarchical Omega and explained common variance, ECV) and came to the estimates for these metrics. Thus, we are confident that our revised scales are more accurate, thanks to the reviewers helpful feedback. Moreover, we use these grouping of questions in other analyses used for Figure 2).

#####

Detailed Comments:---

Page 2 Line 24: Strictly, "rival" should be "rivalrous." (Or, "resources that rival each other...but are non-excludable..." Or maybe, "...resources that are rivals...but are non-excludable...")

#####

Reply 8: While the reviewer is grammatically correct, we maintained the term because a "rival good" is economic jargon for a good that can only be consumed by a single user.

#####

Page 2 Line 42--43: for [the] ecological functioning. Think you need to add the definite article.

Page 2 Line 43--50: The two "lists" in this paragraph are ambiguous. The "from" that starts the first list does not apply to all of its elements except if you itemize/enumerate them. If not, then you should use "from provisioning ecosystem services...from regulating ecosystem service...and from cultural ecosystem services...." More important is the second list, where there is an unclear contrast between the first three items and the "or" that separates the last item. Was this really an "either/or" quantification, or do you perhaps mean, "...the grazing, burning, and ploughing (cultivation) of wetlands, and whether..."? Surely all four activities were quantified?

Page 2 Line 51--52: First instance of "of" should be "on". For the latter, the complete statement would (and perhaps should) be, "information on the demographic data of respondents."

Page 2 Line 53: You need to state your rationale. "The reason for collecting demographic data is that...."

Page 3 Line 6: "respondents" needs a plural apostrophe (respondents') and a comma after "community". But the following might be better: "...the questionnaire asked respondents about their roles in the community, to distinguish...."

Page 3 Line 15: Should be, "The information on...."

Page 3 Line 34--36: Meaning of the last sentence is not clear; presumably it's that, because of the area's uniqueness, a loss of biodiversity would have a global impact. If so, then try, "..., MAP is regarded as a geographic[al]* area where a local loss of biodiversity would have a global impact." [* Note: Figure 1 uses "geographical" not "geographic".]

Page 4 Line 12--13: "..., which we assumed to be an underlying cause..." would be better.

Page 4 Line 24: Need "into Xhosa".

Page 4 Line 37: Possessive case: needs to be, "respondents' demographic variables...."

Page 4 Line 48--49: Incomplete. Try, "The even-numbered questions of Section B were used to determine [assess] damage-causing activities...."

#####

Reply 9: All these changes have been implemented in the manuscript and are highlighted for easy reference.

#####

Page 4 Para beginning Line 50--51: Based on an examination of your code, I don't see that your description matches what you did. For instance, for the `benefits.self ~ benefits.group` regression, the regressor and regressand share two questions (Q5, Q11), the other three being different (so five questions each); whereas for the `damage.self ~ damage.group` regression, the regressor and regressand are each based on four questions (so four questions each), with no shared questions. Clearly you made a mistake when calculating mean-scores for the `benefits.self` and `benefits.group` subscales. But even then, your description remains confusing and confused. It's clear to me, but only because I scrutinized your code and the accompanying analyses, that each scale is based on four questions, each of which was asked twice, giving eight sets of answers for each scale, and a total of 16 sets of answers for the two scales together. It is misleading to say, as you do, that each question in section B was asked twice, because in the preceding paragraph (Line 41) you say that "Section B comprised of 16 questions." If each question was asked twice, and there are 16 questions in all, then there should be 32 sets of answers or scores, which there are not. Note also that sometimes you capitalize "Section B" in mid-sentence and sometimes you don't, writing "section B."

Page 4 Line 52--53: Leave out "asking."

Page 4 Line 54: Should be "are," not "is" [to see this, leave out "perceived"].

#####

Reply 10: We rewrote this section for clarity and ensured that we used correct capitalisation throughout the manuscript. Moreover, we are now sure that question responses were correctly assigned to self- and group-assessments (available in the R-code on GitHub). The grammar issues have also been corrected.

#####

Page 4 Line 56 & Page 5 Line 6--9: See point 2 of my introductory remarks for discussion of the problems associated with these regressions and with your interpretation of them. Briefly, addressing just the core problem, you fitted no-intercept models, which fundamentally changes the interpretation of R^2 , the coefficient of determination. Notably, it loses all connexion to the correlation-based interpretation that you give it. And just as significantly, you omitted to tell the reader that this is what you did. The R^2 for the intercept-based benefits model is 0.01964, which is equal to $\text{cor}(y,x)^2$, which in turn is equal to a Pearson correlation of 0.1401. In short, this means that there is no demonstrable relationship between the two sets of scores, as measured by Pearson correlation (or, in fact, Spearman/Kendall correlations). Yet R^2 for the no-intercept version of the model is 0.944. An informed reader, unaware that you've fitted a model without an intercept (because nowhere do you say that that is what you fitted), would assume that a strong linear relationship between the two forms of the benefits variable exists (Pearson correlation = 0.972), when in fact this is not so. The same general remarks apply to the other regression. The results you present in Table 2, and in the second paragraph of Results on pages 5 and 6, are therefore not only wrong but highly misleading.

#####

Reply 11: As mentioned earlier, these analyses have been corrected based on the reviewers feedback and Table 2 has been removed

#####

Page 5 Line 17: Should be "odd- and even-numbered...."

Page 5 Line 32--34: The following is not true: "...every single respondent was Xhosa, meaning that there was no language variation within the community." Your dataset shows that one respondent spoke seSotho and one spoke English. They may all have identified themselves as belonging to the Xhosa people, but that is an essentially different thing.

#####

Reply 12: These have been corrected in the text and highlighted in yellow.

#####

Page 5 Line 35--37 et seq.: Pairwise correlation is not a recommended/accepted method of testing for

(multi)collinearity. See point 3 of my introductory remarks.
Page 5 Line 36: Should be "correlated with".

Reply 13: This is not longer relevant as the analyses have changed and these sections have been deleted
#####

Page 5 Line 43--45: Results are selective because there are no grounds on the basis of collinearity for excluding the predictors you did. Also, you need to tell the reader that the F-values and P-values you report in Table IV are based on a sequential decomposition of the sums of squares, i.e. so-called Type I or sequential ANOVA, rather than on Type II or Type III (which here are the same).
Page 5 Line 51--52: "...and low levels of education..." is better (education is a noun, not an adjective).

Reply 14: Corrected and highlighted in the text.
#####

Page 5 Line 52--55: The statistics you report are highly misleading and do not support your claims. As I've noted above, and in my introductory remarks, R^2 from a no-intercept regression does not measure association. To get that type of interpretation you need to fit a regression model with an intercept term. The square-root of the R^2 value from such a regression would indicate the degree of linear association and for your data is $r = 0.140$. The R^2 and beta (slope-coefficient) for the model are 0.020 ($r = 0.140 \sim 0.020^{.5}$) and 0.100, respectively.

Page 6 Line 39--43: This is the same issue as that just noted, namely interpreting the statistics from a no-intercept model as if it's an intercept-based model. Categorically, the R^2 value from a no-intercept model does not estimate the amount of variance in the response variable explained by the predictor(s). See point 2 of my introductory comments and the references I cite there. If one fits a model with an intercept term then the variance in self-reported damage that the model explains is 36% not 75%, and the slope-coefficient is 0.652 not 0.528. Hence, respondents are not twice as likely to attribute damage to someone else than to themselves. Pearson's correlation coefficient is 0.584.

Page 6 Table 2 Caption: Somewhat muddled, and you have written "from from" (repeated). Maybe try something along the lines of: "Summary statistics from separate linear regressions of self-reported benefits and self-reported damage on perceived community benefits and perceived community damage, respectively."

Page 6 Table 2: The statistics you report do not have the interpretation you have given them. As I have noted above, you interpret these statistics as if you fitted models with an intercept term but the truth is that you fitted no-intercept models while omitting to say so. Further, for the no-intercept model for Benefits, the slope-coefficient you report should be 0.883 not 0.880.

Page 6 Line 55--56: "...wetland, results found a positive, statistically significant relationship..." Results themselves do not find, they suggest or indicate some result or finding. You might say, "...wetland, we found..." or "...wetland, the results indicate..." or "...wetland, the results support the finding/fact..."

Page 7 Line 30--32: You fitted an additive model (Table 4, Figure 3), which conditions on, or takes account of, the effect of other predictors in the model; it does not test whether the effect of self-reported damage on self-reported benefits (Figure 3-a) depends on values (or levels) of these other predictors. Hence, you provide no evidence to support your claim that the positive effect of self-reported damage on self-reported benefits is consistent across age, gender, level of education, and employment category. To test this, you would need to have fitted a multiplicative model in which self-reported damage interacted with the other predictors, i.e. a model of the form: $\text{benefit} \sim \text{damage} * (\text{other predictors})$. If you do this, then your claim indeed holds good; however, nowhere do you explicitly fit, discuss, or present the results of such a model.

Page 7 Line 34: Residency is not collinear if one uses a reliable test to test collinearity.

Page 7 Line 36--38 & Table 3: I am not convinced by this approach, which strictly speaking is wrong. The main problem is that you are taking means and standard errors of individual questions as if each is a continuous variables, which it is not. Each is an ordinal-level variable. While most authorities consider it to be permissible to take the mean of several ordinal-level items/questions and to treat the result as a continuous variable, this is not true for the analysis of individual questions. Each of your questions is an ordered factor with just three levels or categories (0,1, and 2), i.e. your questions are not an interval- or ratio-scale variables, which would allow the calculation of such statistics without criticism. The problem is essentially the same as one I mentioned above: when using Revelle's omega function to calculate statistics on reliability and dimensionality for your scales (to measure benefits and damage)

you used Pearson correlation (the default) rather than polychoric correlations. See Carifio & Perla (2007), Brown (2011), and elsewhere on this issue. I also don't think that you have made it clear enough to the reader that these four activities represent the measurement variables of the scale you used to measure damage to the wetland, i.e. the eight even-numbered questions of your survey. Finally, if you were to use this type of approach (but you should not) then you need to use confidence intervals (bootstrapped) not standard errors. See §3 of my report where I suggest replacement analyses. For the Wilcoxon signed-rank tests that I carry out there, all you would need are the estimate of the location shift (Est. = pseudomedian), the associated 95% confidence intervals, and perhaps p-values (for exact p-values in the presence of ties see wilcoxsign_test() in package coin). Arguably a more powerful/detailed approach would be to use Cliff's delta statistics. See R package orddom.

Page 7 Line 36--38: To make this clear, you need to add "self-reported damage" or "Damage-causing activities" in brackets after "...they partook grazing, ploughing, dumping or burning" so that there is an explicit link to Table 4/Figure 3-a.

Page 7 Line 38: Preposition missing: usually one "partakes of" or "partook in." But the real problem here is that it is not the residents/respondents themselves who are doing the grazing, it's the livestock. You are looking for something simple like, "...was how often they used the wetland for grazing, ploughing, dumping or burning."

#####

Reply 15: All the aforementioned is no longer relevant as the analyses have changed and these sections have been deleted

#####

Reviewer: 2

Comments to the Author(s)

Nicely written paper but small sample size. Large sample size (>350) may affect your result/inference.

#####

Reply 16: It is not possible to implement this suggestion because there are only 150 households in the region. Our sample size 33% was representative of the community as a whole.

#####

Reviewer: 3

Comments to the Author(s)

The study is important as it is from a region with limited literature on community dependence on wetlands, thus it is an area of interest. However there are certain flaws that cannot be ignored. Hence this manuscript cannot be published in its present form.

Detailed comments are as:

1. The manuscript needs proper editing.
2. Introduction is fluid and lacks conviction

#####

Reply 16: Large section of this manuscript have been rewritten to accommodate the changes to analysis, so we believe that the editing and conviction of these reworked sections is of a higher standard than the original submission.

#####

3. The title reads as "Ecosystem services and ecological degradation of communal wetlands in a South African biodiversity hotspot", but the study reported in the manuscript is from a single village located in the biodiversity hotspot and the study also reflects the results from a single wetland, hence there is a complete mis-match of the study design and the title.

#####

Reply 17: We did not change the title of this revised manuscript because it does reflect the scope of the study, which is the ecosystem service and ecological degradation. Moreover, the abstract makes it clear that it is a survey of 50 households from a single administrative area. It should be noted, however, that the region does not contain only one wetland. Instead, the national wetland database identifies 231 wetlands in the administrative area. Unfortunately, there are known errors in this data base, so we revised the section on the study area to make it clear that there are "more than 200 wetlands" in the region.

#####

4. The study area description fails to provide the overview on the area and type of wetland and the ecosystem services provided therein by the wetland. It is crucial to enumerate few of the ecosystem services and goods provided by the particular wetland, which is the focus of the study.

Reply 18: Since it was not the intention of this manuscript to conduct an ecosystem services assessment (but rather to evaluate the self-reported benefits from wetlands) we did not feel it necessary to quantify all the wetlands in the region. Instead, we focused on how often residents benefitted from provisioning services (water and vegetation), regulating services (flood attenuation and sediment retention) and cultural services. This is described in lines 113-127.
#####

5. It will be useful to know the species assemblage in Ward 32 where the study has been conducted. Particularly interesting will be the name of some endemic species of the region

6. Page 4; Lines 6-8: The authors have mentioned Protected Areas only once in the entire manuscript. However it will be interesting to know the percentage coverage of the MAP under protected area network.

Reply 19: While we agree that these are pertinent points, we are of the view that this information is tangential to the rest of the manuscript. Therefore, we have included a citation to a paper in line 82 that contains an assessment of biodiversity patterns in the MAP biodiversity hotspot for reader who would like to read more.

With regard to protected area coverage, this too is a specialised, but tangential, topic for this manuscript. The trouble with this issue is twofold: first, defining a protected area is quite challenging because the word database for protected areas (i.e. the source used for tracking targets towards the Conventional of Biological Diversity Aichi targets) differs from the South African protected area database (because the national protected areas act distinguishes between degrees of protection); second, most audits of protected areas are either at the scale of ecosystems (i.e. biomes or ecoregions) or at administrative areas (i.e. national or provincial boundaries). Therefore, there is no source on the protected areas levels of the MAP hotspot. We, therefore, quantified this ourselves using various protected area database and found that 2.65-6.69% of the hotspot is formally protected (depending on which database we use). To be conservative, we state in line 84 that less than 7% of the MAP biodiversity hotspot is formally protected.
#####

7. Page 4; lines 9-10: It will be useful to provide a list of traditional activities carried out by people in the wetlands.

Reply 20: In lines 255-267 we briefly mention how cultural affects the use of ecosystems by different genders, which includes a reference [37] that elaborates on the nature-based traditional activities by Xhosas.
#####

8. What kind of environment degradation the authors are referring to? The authors need to specify.

Reply 21: This is specified in lines 128-129.
#####

9. Page 4; Lines 15-17: "Furthermore, each of the 56% and the youth unemployment ratio was 61%."
Replace ratio with percent.

Reply 22: This is corrected and highlighted in the manuscript.
#####

10. Page 4: Questionnaire Design

This section needs to be re-written to reflect the variables used, the type of questions used (open-ended or close-ended). If close ended what was scale used?

#####

Reply 23: A detailed explanation for the scale used in close-ended questions is provided in lines 139-147 of this revised manuscript.

#####

“Differences between these categories were tested using a repeated measures ANOVA (lme command in nlme package) followed by a post-hoc multiple comparisons pairwise t-test with Bonferroni correction for multiple comparison”

Do the author mean that difference between responses to the ecosystem services were tested? The authors need to rewrite this section to make the design clear.

Reading the results and specifically Figure 2 show that the authors have tried the t-test between the two groups of respondents: one which always derived benefits and the other group which occasionally derives benefits. This analysis doesn’t make any sense. What is the scale used in figure 2 (Y-Axis)? It is unclear.

#####

Reply 24: The methods referred to here have been changed to non-parametric alternatives as suggested by Reviewer 1 and the methods section has been rewritten to explain these changes (lines 120-127).

Regarding the figure, which is now Figure 3 in this revision, we acknowledge that the figure caption was unclear and this led to the reviewer misinterpreting the plot. The analysis is not comparing groups of respondents, but rather comparing how frequently respondents reported that they benefitted from the three classes of ecosystem services presented here (provisioning, regulating and cultural). We have, therefore, reworked the caption to figure 3 to make it more clear what was being compared.

#####

The methodology is flawed and the authors need to rework on the explanation of questionnaire design and analysis. I suggest that they refer to some previously published papers on wetland benefits and community attitude (Ambastha et al., 2008; Hussain and Badola, 2008, 2009; Badola et al. 2013).

#####

Reply 25: We respectfully disagree that our methods are flawed, while acknowledging that no questionnaire is ever perfect. Unless the reviewer can provide detailed remarks besides those we have addressed above, then we will assume that our study design is technically sound for the scope of this study. With regard to the additional resources, we could not find these papers as there is insufficient bibliographic information. However, after exploring the reviewer’s publication record, we did find two papers that we felt were relevant to this study (cited as sources 34 and 35 in our manuscript).

#####

11. Page 9; lines 35-39: The authors have justly used the Ostrom’s concept of sustainable use in the discussion but have failed to reflect the same in their methods and results.

#####

Reply 26: We did not elaborate on Ostrom’s work in the introduction and methods because we are not testing her ideas, but rather using her ideas to interpret our results. Therefore, we maintain that the detailed discussion of her work is best suited for the discussion section. Nevertheless, the concept of tragedy of commons and the importance of collective-choice rules is introduced in lines 20-29 of the introduction.

#####

Appendix B

Dear editors,

Thank you for reconsidering your original decision and allowing us to resubmit this revised manuscript to Royal Society Open Science.

We are sympathetic of the difficulties finding reviewers for the many submissions at your broad-scope, open access journal. So, we certainly do not want to exhaust your pool of reviewers.

As we mentioned in our appeal letter, we believe that the referees' remarks could be addressed with minor modification to the manuscript. We outline in detail below how we dealt with each of the referees' comments, but briefly provide a summary of how these issues have been addressed here.

The first referee (Reviewer 4) seemed to be most concerned about the flow of information and the continuity between sections. We agree that the introduction could be strengthened by subtle changes, which we have implemented in this revision. However, we disagree with the referee's suggestions for possible structural frameworks and we explain in detail below why our approach is a more suitable alternative. The rest of the referee's remarks were minor amendments in the text, which we implemented as requested.

The second referee (Reviewer 2) had a more substantial criticism that we did not quantify actual ecosystem services, but only people's perceptions of them. This is a valid criticism, but not one that affects the outcomes of our study. We explain below (with reference to other studies) why this is not a critical oversight and why it will not affect our results or conclusion. Other remarks by the referee were either minor content issues (which were corrected) or requests for tangential information (which we explain below are unnecessary).

Most importantly, we would like to emphasise that neither referee had any substantive criticisms about the methods used in this study, the interpretation of the data, or the conclusions drawn from the results. Of course, they may have mentioned this in their private comments to the editors, but without access to these remarks, we see no cause to make comprehensive changes to this manuscript.

Lastly, both referees' mentioned that the writing in this manuscript was sub-par. We realise this and have revised the language and grammar of the whole manuscript. However, since there were only a few specific remarks, it is difficult for us to determine whether we have addressed the referees' reservations or not. Nevertheless, although we are the first to admit that our writing can be improved, we are convinced that the main results and conclusions of our study will be clear to readers.

In order to make this editorial process as simple as possible, we have highlighted all changes to the manuscript in yellow, and we provide a point by point description of these changes in the response letter below (including specific references to lines in the manuscript).

We hope that you find our amendments adequate and we keenly await your decision.

Kind regards,

Falko Buschke

What follows is a point-by-point reply to the reviewers comments.

#####

Associate Editor Comments:

While recognising you have worked hard to address the concerns of the first round of review, the referees continue to have issues with your work, and do not feel able to recommend it for publication. As the journal's policy does not generally allow for multiple rounds of revision, we regret that we cannot consider the paper further for publication. Nevertheless, many thanks for considering Royal Society Open Science for your work, and we hope the referees' comments are valuable if you choose to submit the paper elsewhere.

***** Reply 1:** We certainly understand the challenge of finding referees for multiple rounds of revision and we do not want to contribute to the widespread reviewer-fatigue that is gripping scientific publishing. We have, therefore, made a concerted effort to not only address the reviewers concerns thoroughly, but also to outline

these changes in this detailed response letter. We hope that this will ensure that making an editorial decision will be as simple and streamlined as possible, without over-taxing any additional handling editors or referees.

#####

Reviewer comments to Author:

Reviewer: 4 Comments to the Author(s)

The manuscript studies human-nature interactions using as a case study the ecological systems of wetlands and a rural community living in a biodiversity hotspot in South Africa as a social system. Interactions are based on benefits derived from the wetlands (ecosystem services) and damage-causing activities affecting the wetlands (ecological degradation). It brings to light: i) the relationship or lack of relationship between damage-causing activities, and benefits, and ii) the demographical factors that influence above interactions. The insights gained through this study are valuable as they cover an understudied, yet vulnerable area. However, the paper would benefit from a restructuring of content around the two enounced research questions. The manuscript went through the first round of revisions, and I acknowledge a lot of work has been invested in the statistical analysis, both from reviewers and authors. A general check for the English language is also recommended. Below are some specific comments.

***** Reply 2:** We acknowledge that our early draft did not present information as concisely as this revision. We agree with the referee that the content should be built around out two main questions. However, we are convinced that clarity can be improved by rearranging the existing information, rather than adding or removing content.

So, in addition to addressing the specific remarks outlined below, we also reworked the introduction to be more accessible to a first-time reader. These changes were subtle, but the effect is noticeable. After considering the referee's remarks, we realised that we had been presenting information in a specific-to-general logical sequence (e.g. In this study we quantified X. Previous research has shown that X is important because of concept Y). By flipping the same narrative to a general-to-specific logical sequence, we are confident that we improved the logical flow (e.g. Previous research showed that concept Y is important. So in this study we quantified X because it represents concept Y). These changes are highlighted in lines 44-45, 46-47 and 49-50.

We also added strategic 'signposting' sentences in the introduction to provide some additional context to the first-time reader. (for example, line 35-36). Lastly, we revised the whole manuscript to sharpen the general language, focussing specifically on using tenses consistently and removing unnecessary adjectives and adverbs. While we realise that evaluating the quality of language and grammar is often subject, we are confident that this revision is clearer than its predecessor.

#####

ABSTRACT

The abstract correctly reflects the content of the paper.

Consider clarifying or separating the following statements for the abstract: "Respondents were more likely to blame others for wetland degradation, although there was no link between the damage-causing activities and benefits from wetlands." Is this reflected as such in the Results section?

***** Reply 3:** We did not modify this sentence in the abstract because we are confident that these statements in the abstract are indeed reflected by the results.

The finding that respondents tend to blame others for degradation is the main pattern in the right panel of Figure 2 and the statistically significance of this pattern is outlined in lines 178-181. The lack of link between damage-causing activities and benefits from wetlands is reflected by the regression coefficient in Table 2, which did not differ significantly from zero. This finding was also described in lines 189-191.

#####

INTRODUCTION

Lines 6-7: it is not clear how the stated goal “This study, therefore, set out to examine whether the people responsible for degrading wetlands are the same set of people who would benefit the most from their conservation” derives from the previous introductory sentences. Maybe delete “therefore” or make the link between motivation and aim clearer. Also, it is less common that the aim is stated so soon in a flow of a manuscript. Consider moving information about the aim of this study in the last paragraph of the introduction.

***** Reply 4:** As also described in Reply 2, we have adjusted parts of the introduction to promote a clearer flow of information. The specific sentence referred to by the referee has been deleted because it was also presented in lines 54-56.

#####

Lines 30-37: parts of this paragraph rather belong to section 2. Methods (a) Study area

***** Reply 5:** In hindsight, we agree with the referee. Sections of this paragraph have been deleted because they are already included in the description of the study site. However, the remaining parts (lines 27-33) are still in the introduction because we believe that it provides important context on: (1) that the area is a biodiversity hotspot as captured by the title, (2) the region is poverty-stricken and there is a high reliance on wetland ecosystem services and (3) the wetlands there are on communal resources and susceptible to the tragedy of the commons. This information offers a segue between the preceding paragraphs into the subsequent paragraphs.

#####

Lines 38-65: most parts of these paragraphs rather belong to section 2. Methods (b) and (c)

***** Reply 6:** Although we respectfully disagree with the referee on this point, we understand why s/he made this remark. As outlined in Reply 2, we have rearranged the content of these paragraphs (now lines 34-52) to first present the general conceptual context and then explain how it is relevant to this specific study. This is much more logical and eases the first-time reader into the study. Our original structure began with the specific strategy used in this manuscript, followed by the broader conceptual context, which, understandably, gave the referee the impression that the information was more suited for the methods section.

The revised section first provides the broader conceptual context and then outlines how we incorporated it into our study. We are confident that this information is needed in the introduction section, because it provides the first-time reader with context on why we designed the questionnaire the way we did and how the result ought to be interpreted.

#####

Line 71: I am not sure “social capital” is a demographic characteristic. Consider adding “whether demographic characteristics and related social factors such as social capital, ...”

***** Reply 7:** The referee is indeed correct. Our revised text (lines 44-46) makes this more clear because we, first, explain that social capital is an important determinant of sustainable resource use; and, second, outline that we assume that older respondents and those who lived in the region longer would have developed more social capital. Thus, we use demographic variable to infer social capital, rather than measuring social capital directly.

#####

General comment Introduction: Consider restructuring the introduction so that each paragraph conveys a (general) message, that only foresees elements of the Discussion without revealing the results. For example, as a suggestion: a paragraph on wetlands in general, a paragraph on benefits/ecosystem services from

wetlands, a paragraph on threats to wetlands in a certain context, (research gap), a paragraph on the aim and research questions.

***** Reply 8:** We agree with the referee’s assessment, so the introduction is more streamlined in this revised version. However, we chose not to follow the suggested logical framework, but are confident that the framework used in this revision reflects the context of the manuscript equally well. There is no single correct way to write a scientific paper, but we optimistic that our approach provides a broad context while also laying the groundwork so that subsequent sections can be interpreted accurately.

Our revised introduction has the following eight paragraphs: (1) wetlands provide important ecosystem service to rural communities. (2) Some of these services are often overlooked. (3) Wetlands are also communal, which can lead to the tragedy of the commons. (4) The tragedy of the commons can be overcome using collective-choice rules. (5) Our study focuses on ecosystem services from communal wetlands in a poverty-stricken biodiversity hotspot. (6) We assessed how community members benefit and damage wetlands. (7) We link these to demographic variables that can be predictors of the effectiveness of collective choice rules. (8) A closing paragraph with the specific research questions.

Therefore, we maintain that the structure of our introduction is logically consistent. We are confident that it describes the conceptual landscape within which our manuscript fits, provides a broad overview of previously published studies, and provides a foundation for subsequent sections.

#####

METHODS

(a) Study area

Move here related material from the introduction.

***** Reply 9:** As outlined in Reply 5, unessential parts of the introduction have been deleted. The site description in the methods section is largely unchanged because we did not move information from the introduction that was not already in the methods section.

#####

Line 89: consider deleting “which we assumed to be an underlying cause of the environmental degradation”

***** Reply 10:** This phrase has been deleted.

#####

(b) Survey design

Move here material concerning the survey that is placed in other sections.

***** Reply 11:** As explained in Replies 5 and 6, we have rearranged the sections of the introduction that could be considered to be better suited for the methods section. This meant that the re-written parts of the introduction seem less out of place and were, therefore, not moved to the methods section.

#####

Line 98: consider replacing “mainly driven” with “facilitated by/through the collaboration with... “

***** Reply 12:** This change has been made in lines 84-85.

#####

Figure 1 caption: consider deleting “where surveys were carried out”.

***** Reply 13:** This change has been made to the caption of Figure 1.

#####

Line 100: Here "Xhosa" appears for the first time. Add a short explanation on how it is related to the case study area.

***** Reply 14:** We have added a sentence (lines 88-90) that provides some context on how Xhosa relates to the study area. There is obviously a much more complicated history of colonisation and resettlement linked to the controversial colonial and apartheid history of Apartheid South Africa. However, this detailed history is probably irrelevant to our revised manuscript.

#####

(c) Questionnaire design

Why are there separate subsections for Survey design and Questionnaire design?

***** Reply 15:** We distinguished between the survey and the questionnaire design because the former refers to the way the questionnaires was administered (i.e. When was the survey carried out? What language was used? Which medium was used to administer the questionnaire? How many respondents were there? Was there informed consent?), while the latter refers to the make-up of the questionnaire itself (i.e. which questions were included? How were responses scaled? Which demographic variable were quantified? How were these questions evaluated and analysed?).

It is possible to merge these sections under a common sub-heading, but this would be a matter of taste and not would a change the results or conclusions.

#####

Line 126: two tests

***** Reply 16:** This typo was corrected in line 115

#####

Lines 160-165: please check the consistency of the names of the seven/six demographic variables with their names in Table 1 and throughout the manuscript. Also, consider rephrasing in this direction: Seven demographic variables were recorded, and six quantified. Language...

***** Reply 17:** This paragraph was edited for clarity in consistency in lines 150-157. The variables are now the same as they are presented in Tables 1 and 2 and in the rest of the manuscript..

#####

Lines 174-177: Limitations of the used methods are mentioned.

***** Reply 18:** Indeed, we mentioned the limits of our analyses in Lines 164-167 based on the first round of review. There are no foolproof statistical approaches and there is always a trade-off between robustness and pragmatism, which we believe we have balanced sufficiently in this revised manuscript.

#####

General comment Methods: Try to make clearer how the three sections of your questionnaire (lines 109-112) answer your two research questions and/or aim.

Data collection and data analysis are satisfactorily distinguished.

I am not an expert in statistics, but the Methods section might benefit from more structure and clarity around which methods and tests were chosen to demonstrate what? E.g.: "In order to... , we applied... in relation to the first research question."

***** Reply 19:** While we understand the referee's suggestion to streamline the methods section to make it more closely linked to the research question, this was difficult to do in practice because many of the analyses did not directly answer the two main research questions. Instead most of the statistical analyses were necessary precursors to confirm that the data were appropriate for the in the multiple regression analysis:

(1) Cronbach's alpha, McDonalds Hierarchical omega and explained common variance (lines 139-143) were necessary to validate that the data obtained from the questionnaire can be synthesised into single measures of benefits and damages to wetlands.

(2) The Wilcoxon test and dependent sample assessment plots (lines 123-127) were necessary to ensure that self-reported responses were reliable, by comparing them to community-reported responses. This was also an essential step to identify any systematic bias in the responses.

(3) The Friedman test and post-hoc tests (lines 110-116) was necessary to ensure that when respondents were reporting the benefits from ecosystem services, we actually knew which services were more or less beneficial.

Each of these steps was necessary to confirm that the data actually reflected what we assumed it represented. These data were then used in a multiple regression analysis and the coefficients from these regressions (Presented in Table 2) were used to actually answer the two main research questions as outlined at the end of the introduction.

#####

RESULTS

Line 190: consider rephrasing "to someone other than themselves". I remark the point made here is nicely discussed in the Discussion section (lines 271-281).

***** Reply 20:** This sentence has been revised in lines 178-180.

#####

Lines 193-194: consider keeping this for discussion only: "which is indicative of potential gender-based power imbalances in the region".

***** Reply 21:** Although the referee is strictly correct here, we believe that keeping this phrase in the results section (lines 183-184) helps the reader translate the coefficients estimated from the regression analysis into real-world meaning and consequences. We are of the view that the benefits of keeping this short phrase outweigh the costs of deviating from academic convention.

#####

Table 2 caption: consider separating in more sentences. Did the authors explain in the manuscript what is meant by the F-score?

***** Reply 22:** We have rephrased the caption slightly to refer only to demographic variables (rather than listing each variable), which made the caption shorter and easier to read.

With regard to the F-score, it was stated in line 163 that we used a F-test associated with an analysis of variance to test the significance of regression coefficients. Since this is a standard statistical null hypothesis test used during ANOVA present in all undergraduate statistics textbooks, we do not feel it is necessary to elaborate on the details of F-tests and F-scores in this manuscript.

#####

General comment Results: Try to keep the same structure around the research questions, the threefold structure of the questionnaire (example: results to the analysis of section A of the questionnaire covering respondents' demographic information actually answer the second research question, etc.).

***** Reply 23:** We disagree with the referee's suggestion for the same reasons outlined in Reply 19. While the referee is indeed correct that the sequential presentation of information is ideal, both research question are actually answered by Table 2. The other results are necessary steps that lead to the headline results in Table 2. Specifically:

- Table 1 reports the descriptive demographic data without any additional analysis. We presented this first because (a) it gives the reader the necessary context of the demographic make-up of the respondents and (b) this is the simplest, unprocessed data that can be interpreted on its own.

- Next, we present in Figure 2 the comparisons between self-reported and community level responses. This step is important to identify any biases in the responses and ensure that any subsequent statistical analyses are based on reliable estimates of benefits and impacts.

- In Figure 3, we show that on average everyone benefits from ecosystem services, but that they are more likely to benefit from regulating ecosystem services. This is necessary because unless the community benefits from ecosystems service, any subsequent demographic segmentation of ecosystem service use would be pointless.

Only once these three sets of results have been established, does it make sense to even consider reading anything into the multiple regression analysis (Table 2). For example, without (i) community heterogeneity, (ii) unbiased responses, and (iii) actual benefits from wetland ecosystem services, the outputs from the regression models would be riddled with statistical artefacts because the underlying data is uninformative.

We, therefore, maintain that the the order in which we presented the results is the most scientifically robust sequence that ensures that the results and subsequent conclusions can be trusted.

#####

DISCUSSION

Lines 231-232: Consider rephrasing and check grammar: "Community tend not to self-organise when natural resources are very abundant or highly depleted [9,28]."

***** Reply 23:** This unfortunate error was corrected in line 220 by using the plural of 'communities' instead.

#####

Line 238: Consider rephrasing and check grammar: "members will not follow rules today when future outcomes are uncertain".

***** Reply 24:** This sentence has been restructured in lines 226-227.

#####

Line 242 suggestion: "... but also of the relationships between different members of the community and of their nature connectedness."

***** Reply 25:** This change has been implemented in lines 230-232.

#####

Line 284: replace "doomed to fail"

***** Reply 26:** We implemented this change in lines 272-273.

#####

Line 323: suggestion for rephrasing: "they tend to attribute lower priority to these issues compared to technical solutions"

***** Reply 27:** This has been implemented in lines 311-313.

#####

Line 325: Consider rephrasing: "Specifically, building on our results we make three recommendations on how to manage the... ". These are recommendations, or a general roadmap, as the subtitle suggests, rather than a fully laid out strategy.

***** Reply 28:** This recommendation has been implemented in lines 315-317.

#####

General comment Discussion: The discussion is generally well written, and its development is consistent with the rest of the manuscript, including the abstract: "Therefore, we describe how strong leadership could nurture a sustainable social-ecological system by integrating ecological information and social empowerment into a multi-level governance system.

***** Reply 29:** We'd like to thank the referee for this final positive remark, and for the effort of making all the other thoughtful suggestions

#####

Reviewer: 2 Comments to the Author(s)

The specific comments to the manuscript are:

This manuscript is about the ecosystem services and ecological degradation of the communal wetlands, but nowhere in the manuscript the authors have provided the specific ecosystem services to the local community of the study area. The authors have framed this manuscript around the perceived benefits to self as well as other. But how did the authors choose just a few of the provisioning (plant and water), regulating (flood and erosion) and cultural services, as mentioned in the questionnaire design.

***** Reply 30:** The referee is correct that we did not actually quantify the ecosystem services supplied by the wetlands. However, this is not a step that is essential for answering the specific research question in our manuscript. We justify this using two main lines of reasoning:

First, in our main analysis (Table 2), the benefits from wetlands are aggregated into a single unidimensional dependent variable. We could do this because we explicitly tested the internal consistency of our survey instrument (line 137-149). What this implies is that even had we added or removed other ecosystem services in the questionnaire, our results would be robust because these additional responses would also be aggregated into the unidimensional variable (assuming they also represent a single survey scale). This is obviously a simplification of the multidimensional relationship humans have with nature, but it is a necessary simplification for a first assessment in a poorly studied region.

Second, it could be argued that ecosystems services derive their value from how much they are valued by beneficiaries. The very definition of ecosystem services are “activities or functions of an ecosystem that provides benefits to humans” (Mace et al. 2012.) and recent conceptualisations refer to “nature’s benefits to people” (Diaz et al 2015 a,b). Thus, human perceptions of benefits is an important component of the concept of ecosystem services. Similarly, the value of ecosystem services is the relative contribution that ecosystems make to support people’s well-being. Valuation is inherently complex because it includes intrinsic, instrumental and relational values (Diaz et al. 2015 a,b). We framed our study as the way societies interact with communal natural resources, which depends on collective choice rules (as prominently illustrated by our introduction and discussion sections). Collective choice rules depend on human perceptions of the values they gain from nature. Therefore, we feel justified in assessing the value of ecosystem services based on individual’s perceptions of the benefits they derive from nature. This approach is not new and has been used by other studies before ours (Costanza et al. 2014). Our use of a questionnaire-based survey instrument is not only appropriate, but also widely used by other ecosystem service researchers.

Both of these arguments support the methods used in our study. We did not claim to perform a comprehensive assessment of all ecosystem services, nor did we claim to carry out a complete valuation of these services. For the purpose of our study, simplified questions about some ecosystem services were sufficient to answer our research questions in a scientific robust way. Our questions referred to very broad services (water, plants, floods and erosion, insect pests and cultural activities), which allowed respondents to interpret the benefits from wetlands in their own way. In this way, we accommodated the many relational values people have with nature, that cannot be assessed by other means (Diaz 2015 a,b)

- Mace et al. 2012. Biodiversity and ecosystem services: a multilayered relationship. Trends in Ecology and Evolution, 27, 19-26
- Costanza et al. 2014. Changes in the global value of ecosystem services. Global Environmental Change, 26, 152-158.
- Diaz et al (2015a) The IPBES Conceptual Framework – connecting nature and people. Current Opinion in Environmental Sustainability, 14, 1-16.
- Diaz et al. (2016b) A Rosetta Stone for nature’s benefits to people. PLoS Biology, 13, e1002040.

#####

What was the reason that most of the respondents were in their 20s and 30s, and how many were leaders (traditional or formal), as in survey design it is mentioned that community survey was conducted with community members and their leadership. Is community leader a form of employment? I don't agree.

***** Reply 31:** We do not know why most of the respondents were aged 18-39. In all likelihood, it reflects the age frequency-distribution of the region. Unfortunately, the South African Statistics Agency does not provide detailed information for this region specifically, but for the whole Eastern Cape province 30% of the population is between 18-39. Considering that we only interviewed one person per household, it seems statistically likely that the head of the household would be in this age group.

With regard to including community leader as a form of employment: the referee is correct in pointing out that this could be defined as a community role instead of a form of employment. However, we chose to rather list it as a form of employment because in South Africa traditional leaders are paid salaries directly by the Government (these salaries are publicly available in the Government Gazette, but their might be other non-disclosed payments as well). Monarchs, such as the King of the amaPondo from our study area, receive approximately US\$ 80 000 per year, while senior traditional leaders will receive approximately US \$17 000 each year. This is a substantial amount of money for a region with more than 50% unemployment. Since we used employment as an indicator of dependence on wetlands, we could not ignore that traditional leaders have an alternative source of income that reduces their dependence on natural resources. Hence our decision to list traditional leader as a form of employment rather than as a role in the community (although community leader will also be included in the ‘Other’ category of Community role)..

#####

To know demographic composition, it is advised to refer to the census report for the area, which will give accurate information on the demographic structure as well as economic and employment status.

***** Reply 32:** As already addressed in the previous reply, this study took place in a deeply rural area that is not listed in the publicly accessible records by the South African Statistics Agency. Moreover, the last national census was conducted in 2011, so we assume that the information cited from the 2016 Integrated Development Plane (lines 74-82) was the most recently available information.

#####

It will be useful to provide a list of traditional activities carried out by people in the wetlands, in the Study Area section.

***** Reply 33:** We addressed this remark during the previous round of revisions. A list of all the traditional activities is beyond the scope of our study because it requires a comprehensive socio-cultural assessment of practices that may be taboo to share with outsiders. However, in lines 256-260 and 288-289 of the discussion, we described some traditional practices and, more importantly, provide citations to work that is more specific to the issue of traditional Xhosa cultural activities.

It is also important to point out that we asked respondents “whether they used wetlands to express their culture”. This allowed them to interpret this in their own personal way, which is an essential step to incorporating relational values between humans and nature (see Reply 30)

#####

Page 4; lines 47-48: “According to the IDP [22], the local community has low levels of education, which we assumed 89 to be an underlying cause of the environmental degradation.”

Low level of education doesn’t necessarily always results in environment degradation. Moreover, education doesn’t grant the person to be environmentally conscious.

***** Reply 34:** Based on this comment and an earlier comment by the other referee (Reply 10), this sentence has been modified to remove references to the link between education and ecological degradation.

#####

Page 6; Line 48: “.....All forms of benefits were accrued more than occasionally....”, delete ‘were’

***** Reply 35:** ‘Were’ was deleted from line 176.

#####

Page 6; lines 53-54: “...The multiple regression model showed that male respondents were more likely to benefit from wetland ecosystem services (Table 2),” Table 2 shows damage causing activities, whereas the sentence says that male respondents were more likely to benefit from wetland ecosystems.

***** Reply 36:** We are uncertain what the referee is referring to here. Table 2 shows the outputs from the regression model with damage-causing activities and demographic variables (including gender) as predictor variables. Since the regression coefficient for gender is positive and statistically different from zero, we can indeed conclude that men benefit from wetland ecosystem services more than women. We do not understand how these results could be interpreted otherwise.

#####

The language is lax and needs proper editing for grammatical and syntax errors.

***** Reply 37:** As already outlined in Reply 2, we have restructured parts of the text to make this revised manuscript more clear and concise. We have also gone over the entire manuscript to improve the grammar and sentence structure. However, our corrections were based on our own efforts to improve readability, so without more detailed information, we cannot be sure if we have addressed the referee's concerns or not.

#####

Appendix C

Dear Professor Padian and members of the editorial board,

Thank you for handling the review of our submission to *Royal Society Open Science*. My coauthor and I are especially grateful for the patience of handling our interdisciplinary manuscript, which is difficult to review by specialist reviewers. Through this rigorous process of review and revision, we are pleased to say that this revised manuscript is considerably better than the version of the manuscript we sent to your journal originally. The reviewers deserve credit for their contributions too.

We have addressed the most recent round of reviewer's comments as carefully as possible (details provided in the point-by-point response at the end of this letter) and have also completed all the additional information as requested: ethical statements, author contributions, conflicts of interests, funding disclosures and data accessibility (the data and R-scripts are deposited on figshare, doi: 10.6084/m9.figshare.7764035.v1). Therefore, we are confident that this manuscript is robust, reliable and ready for publication.

What follows below is a detailed description of all the changes made to this most recent manuscript. These changes are also highlighted in the manuscript in yellow. We hope that this version adequately addresses the lingering concerns and we look forward to seeing this manuscript published.

Your sincerely,

Falko Buschke

University of the Free State, South Africa.

Specific point-by-point responses

Associate Editor Comments to Author:

Please accept our apologies for the delay in completing review of your manuscript: the original reviewers were unable to assist, and it has taken some time to secure the advice of the reviewer here (though we're grateful for all the support of the reviewers).

The comments of the referee are broadly positive but the Editors want to emphasise two points in particular to the authors:

- 1. Regarding your dataset/code: you indicate you will make the material available via FigShare on acceptance. As the Royal Society Publishing provides this service free for authors' electronic supplements, please ensure these are included in your revision. This is an absolute requirement of publication;*
- 2. The Editors do not consider the referee's comments to preclude publication, but you MUST respond to their comments. You should include the changes and modifications requested, and also provide full responses in your reply to reviewers.*

***** Reply 1:** We understand the difficulty of securing reviewers for interdisciplinary manuscripts, which is why we are glad that the reviewers' insights have improved our submission and made it more suitable for publication.

We have deposited our raw data and R-scripts needed to recreate the figures and tables from our submission on the online repository figshare. These data are available under the permanent DOI: 10.6084/m9.figshare.7764035.v1.

We have also addressed every one of the reviewer's comments below.

Reviewer comments to Author: Reviewer: 5

GENERAL COMMENTS

I must preface my review by noting that I am not experienced in questionnaire surveys and their analysis and interpretation, and therefore I am very limited in terms of commenting on any methodological issues in the paper. Instead I comment mainly from the perspective of my experience in assessing wetland ecological degradation and ecosystem services provision.

The manuscript addresses a relevant and important topic for which there has not been much research conducted in South Africa. From my perspective (and with my limitations in terms of questionnaire surveys)

the study appears to be generally scientifically sound, well conceptualized and implemented, and well presented, and makes a useful scientific contribution with practical application. However, there are three important general issues which should be addressed, as well a few specific issues which require attention, which are given below.

***** Reply 2:** We would like to thank the reviewer for the helpful comments. We also appreciate the reviewer's frankness about not being familiar with questionnaire-based studies like ours. Although we are confident about the technical soundness of our methods (based on the earlier rounds of rigorous peer review), we believe it is important that our findings reach an audience of wetland managers beyond those narrowly focussed on questionnaire-based studies. Therefore, we appreciate the reviewer's insights and we hope that our manuscript is a convincing and reliable source of information to general wetland practitioners.

Elaboration on damage to wetlands and the basis on which specific activities are deemed to be damaging or not: Although the paper identifies four damage-causing activities fairly early on in the paper (Grazing, Ploughing, Dumping and Burning) it lacks any specific indication of what is meant by damage to wetlands and the basis on which specific activities are deemed to be damaging or not. It is important that this be done given that damage-causing activities are a central focus of the paper. The implication seems to be that all of the four listed uses are inherently damaging, and while this probably stands for ploughing and dumping, it is more nuanced for grazing and burning. Wetlands over much of South Africa, and probably in the study area, evolved under fire and indigenous grazers (e.g. buffalo) (Fynn et al. 2015 and Kotze 2013) and where large grazers such as buffalo are no longer present in an area then domestic livestock can to fair degree replace their effect. The damage resulting from fire and grazing relates to the specifics of the grazing/burning regime, e.g. from annual fires or prolonged intense grazing.

Fynn RWS, Murray-Hudson M, Dhliwayo M, Scholte P, 2015. African wetlands and their seasonal use by wild and domestic herbivores. Wetlands, Ecology and Management 23:559–581

Kotze D C, 2013. The effects of fire on wetland structure and functioning. African Journal of Aquatic Science 38: 237–247

***** Reply 3:** We agree whole-heartedly with the reviewer about the importance of grazing and fire in maintaining ecological functioning in wetlands (as well as other ecosystems, like grasslands). That said, our study area lacks rotational grazing practices and a formal burning regime. Although we must acknowledge that we have not actually quantified the detrimental effects of grazing and burning on wetland functioning, we believe that the balance of evidence from earlier studies in the wider region suggests that chronic overgrazing and frequent burning has lead to wetland degradation (though defoliation, sediment mobilisation, soil compaction and increased water turbidity). This rationale is now elaborated on in lines 120-124, which specific reference to previously published papers:

“Although ploughing and dumping waste are clearly detrimental to wetland integrity, historical grazing by wild ungulates, contemporary grazing by livestock and natural burning regimes can be important drivers of wetland functioning [25,26]. However, lack of rotational grazing and inadequate fire management in the region has lead to chronic degradation of the wider landscape [27,28].”

A greater depth of reporting on the different wetland uses/benefits: A greater depth of reporting is required on the different uses made of the wetlands in the study area, the compatibility/incompatibility of these respective uses and how demographics (in particular gender) may be specific to particular uses. This would help given the reader more insights into some of the key findings reported in the paper, e.g. that males were more likely to benefit from the wetland than females. Also see specific comments on Page 7, Figure 3.

***** Reply 4:** This suggestion of a trade-off analysis and a subgroup analysis is an interesting one and it would definitely provide valuable insights if it could be carried out appropriately. However, we are reluctant to elaborate too much into the uses of wetlands for three reasons:

First, our questionnaire was very simple by design in order to avoid miscommunication during translation and oral administering of the survey. Therefore, it is not possible to quantify the extent of the damage causing activity (e.g. our questionnaire did not distinguish between respondents who allowed an entire herd of cattle to graze the wetland and respondents who only had one or two cows), which would be necessary for evaluating potential trade-offs between benefits and damage caused to wetlands. For instance, it might be

that moderate grazing is compatible with the continued supply of ecosystem services like clean water, but that heavy overgrazing might not be.

Second, our results only report on the perceptions of respondents, not actual empirical evidence of benefits and damages. Therefore, the gender differences we report in Table 2 might represent differences in how men and women perceive benefits, rather than the benefits they actually receive. Understanding this subtlety will, however, require additional research and we do not want to speculate too widely based on our dataset.

Third, we are especially reluctant to perform a subgroup analysis on gender because this would constitute a form of data dredging where the same dataset used to identify the hypothesis is subsequently used to test the very same hypothesis. Not only would this increase the Type I error rate due to multiple comparisons, it further reduces the statistical power of our dataset by reducing the effective sample size of each subgroup.

While it is possible to perform exploratory analyses without carrying out inferential statistics (see below), we do not feel that this would be appropriate for the main manuscript. Moreover, these exploratory analyses suggest that males generally benefit from provisioning and cultural services more than women, but there is substantial overlap that makes subgroup interpretations ambiguous. If readers would like to explore these subgroup analyses themselves, then they can use the raw data which is publically available.

Review Figure 1: Exploratory comparison of how respondents of different genders benefit from wetland ecosystem services. Boxplots show the minimum, 25th percentile, median, 75th percentile and maximum values (points reflect outliers). Here, 0 denotes that respondents *never* benefit, 1 that they *sometimes* benefit and 2 denotes that they *always* benefit from wetlands.

Reporting on collective choice rules for wetlands in the study area:

The paper makes several general references to collective choice rules and has a sub-section in the discussion on Governance. One would therefore anticipate that some specifics on these rules would be reported on for wetlands in the study area. But this seems to be entirely missing. While acknowledging that this was not the focus of the questionnaire, it is relevant to the study objectives and deserves some attention, even if it is that no collective choice rules for wetlands in the study area appear to exist.

***** Reply 5:** It is unclear to us what exactly is being requested here. Collective choice rules would still need to be negotiated by local community members, so it is not feasible for us to describe these rules in this manuscript. However, we feel it is necessary to state that general wetland management best-practice already exists in South Africa, so the collective choice rules would focus on the *implementation* of these rules (while indigenous knowledge could inform management best-practice, the main aim of local decision-making

is less about 'what' should be done, but rather 'how' it should be done). We have made this case in lines 288-292.

SPECIFIC COMMENTS

Abstract, "However, wetlands are often on communal land and face degradation by individuals who maximise their personal benefit from ecosystem services without bearing the full environmental costs of their actions." I suggest deleting this sentence, which is general and I am also not sure fully substantiated. This would provide more words available in the Abstract for the authors to report on an additional specific finding of the research in place of this general statement.

***** Reply 6:** We are of the view that this sentence is important in the abstract to explain the challenge of managing communal resources. That said, the original sentence was rather clunky, so we have rewritten it for clarity.

Page 2, line 15-17, "Common-pool resources are resources that are rival (i.e. 16 using the resources reduces its availability to other users), but non-excludable (i.e. one cannot 17 restrict access to these resources)" It is not necessarily so that common-pool resources are non-excludable, as in fact is identified in the following paragraph's reference to collective choice rules (which may include exclusions).

***** Reply 7:** This is indeed a subtlety that was missing. Common pool resources can be exclusionary, but only if all the users agree to this exclusion (i.e. no single individual has the right to unilaterally exclude other users). We made this subtly more explicit in lines 16 and 17.

Page 3, line 72, "These wetlands are....prone to degradation caused by man-made activities." /A few words indicating why they are prone to such degradation would be useful

***** Reply 8:** We have clarified in lines 74 and 75 that these wetlands are prone to degradation because there are no restrictions on who can use these ecosystems.

Page 4 line 100-104. "(2) how often respondents partook in the most common damage-causing activities to wetlands (identified during prior site visits). A few words should be added in terms of what was carried out in the "prior site visits". As it stands, this is not clear.

***** Reply 9:** This has been clarified in line 103. The earlier site visit was just an informal reconnaissance trip without any formal quantification of impacts. Instead, we just observed ploughing, grazing, burning and dumping, which were subsequently added to our questionnaire.

Page 6, line 189, "Notably, respondents who were more likely to partake in grazing, ploughing, dumping or burning did not stand to benefit any more from wetland ecosystem services than their peers." Sorry this is a little unclear. Are the "peers" being referred to here not likely to partake in grazing, ploughing, dumping or burning? This should be clarified.

***** Reply 10:** This sentence is based on the non-significant regression coefficient in Table 2. From Figure 2, it can be inferred that both self-reported and group-perceived damage happens slightly less than 'Occasionally' (the vertical and horizontal grey dashed lines are slightly less than 1). However, the newly clarified sentence in lines 195-197 is referring to the result that respondents who partook in damaging activities more frequently did not also benefit more from wetland ecosystem services.

Page 7, para 1: "perceive group" should read "perceived group"

***** Reply 11:** Corrected in the caption for Figure 2.

Page 7, Figure 3: In the text accompanying Figure 3 it would be informative to report on some of the specific services within the three broad categories reported in /Figure 3, namely provisioning, regulating and cultural services. As it stands I cannot see it reported anywhere in the paper.

***** Reply 12:** This information on how the specific responses of benefits were categorised into the three ecosystem service categories in Figure 3 was reported in the methods section (lines 109-111).

Page 8, line 193, "This study showed that all community members benefited from ecosystem services from wetlands in the Hlabathi administrative area, South Africa," This statement could be better substantiated through more specific reporting.

***** Reply 13:** There are several different lines of evidence to justify this statement. First, in Figure 2 the mean benefits (dashed grey lines) show that both individuals and the group benefit from wetlands more than *occasionally*. Second, in Figure 3, respondents benefit from all the ecosystem service categories more than *occasionally*. Finally, the intercept coefficient in Table 2 (1.506) implies that, on average, all respondents benefit from wetlands somewhere between *Occasionally* and *Always*. This is stated explicitly in the results section (lines 182-184). Therefore, we think the general claim made at the start of the discussion section is wholly supported by the data.

Page 8, line 193-195, "This study showed that all community members benefited from ecosystem services from wetlands in the Hlabathi administrative area, South Africa, but men tended to benefit more than women." This statement needs to be substantiated and elaborated on with more specific reporting, in particular giving the reader a better sense of which of the specific services this applied to, thereby helping the reader to better understand why men tended to benefit more than women.

***** Reply 14:** As already explained in Reply 4, we are reluctant to speculate why men reported more benefits than women. Based on our analysis, the significant regression coefficient in Table 2 confirms the difference between men and women. Unfortunately, we believe that elaborating on the deeper mechanism for this gender imbalance would be drawing unjustifiable conclusions from our data. In our original submission, as in this version, we highlight the importance of interrogating further the underlying reasons for our reported gender discrepancies (lines 260-267).

Page 8, line 197-199, "Moreover, there was no evidence that the men and women who degraded wetlands gained more from wetland ecosystem services than their fellow community members." More information is required in terms of what constitutes someone who degraded wetlands. See General comments.

***** Reply 15:** This issue has already been addressed in Reply 10

Page 8, line 216: I am not sure that "seasonal grazing" is a good example of a resource which is mobile, which seems to be implied.

***** Reply 16:** This has been clarified in lines 222 and 223, which states that the grazing lands themselves are not mobile, but that they require different usage rules during different times of the year.

Page 9 line 218-219, "However, the productivity and predictability of these ecosystems might be potential obstacles to their sustainable use." This statement needs to be better elaborated upon, e.g. productivity and predictability in terms of what?

***** Reply 17:** This has been clarified in lines 225 and 226 to explain that the productivity and predictability of ecosystem services can affect how communal wetlands can be sustainably utilised.

Page 9 line 242-244, "Community members have a shared interest in preserving wetland systems, which frees them from the conflict caused by different land-use pressures (e.g. [35,36])." Sorry, it is unclear to me while a general shared interest in preserving wetlands should free community members from conflict caused by different land-use pressures. I can appreciate how a shared interest could help in resolving conflicts, but it seems unrealistic that it would free users from conflicts.

***** Reply 18:** This has been clarified in lines 250-251. If everyone benefits from wetland services, then there is less conflict than is the case when some community member benefit and others do not (as is reported in the cited papers, which were requested by the referee during the first round of review)

Page 9 line 248, "Gender is an important determinant of pro-environmental behaviour" Some elaboration of why this should be so would be useful. In addition, what would constitute pro-environmental behaviour?

***** Reply 19:** This has been elaborated on in lines 255-256.

Page 10 line 279, "The final predictor of sustainable use of common-pool resources is good governance structures with collective choice rules" As elaborated on in the General comments, the paper provides important general statements such as this but seems to be lacking in terms of reporting on the specifics of any collective choice rules for use of the wetlands in the study area.

***** Reply 20:** This point has already been addressed in Reply 5 above. The collective choice rules will need to be negotiated by local users, so it is not our role to impose these rules in this manuscript.

Page 10 line 286, "This is not because respondents are generally selfish or have negative perceptions of nature. On the contrary, respondents were Xhosa, a cultural group that has been shown previously to score highly on various measures of nature connectedness [27,39]. Such human-biosphere connectedness is central to sustainable use of ecosystems and resilient social-ecological systems [48,49]." Some of the assumptions in these statements are potentially problematic, in particular what appears to be an assumed overriding influence that so-called "nature connectedness" might have over immediate self interests.

***** Reply 21:** We respectfully disagree with this suggestion because the very next sentence in this paragraph (lines 301-302) elaborates that the challenge is translating nature-connectedness to broader society and its institutions. The reviewer is correct that nature-connectedness needs to overcome incentives for self-interest, which is exactly what is being conveyed by our statement that such connectedness needs to be reflected in the institutions (formal and informal, which includes incentives) that govern communal resources.

Page 10 line 292, "It is possible that communicating scientific information to traditional leadership structures could establish sustainable practices throughout the wider community." What appears to be the implicit assumption here may often not hold that altered awareness/perceptions will necessarily lead to altered practices.

***** Reply 22:** This comment is indeed true and it relates to Reply 5 above. The collective choice rules are primarily about implementation of sustainable practices. This will only arise through a deliberative negotiation between resources users. Traditional leaders do tend to have more influence, so they are valuable conduits for scientific information throughout the rest of the community

Page 10 line 299 "Instead, it needs to be embedded into a multi-level governance structure (e.g. [52]) that includes national initiatives, like the Working for Wetlands expanded public works programme" I am not sure that a public works programme is the best example to give for a natural resources governance structure.

***** Reply 23:** We respectfully want to point out here that the reviewers reservations are based on an incomplete reading of the first part of this sentence. The first part of the sentence states that public works programmes are important for sustainable wetland usage, but the second part of the sentence (lines 312-313) makes it clear that these formal government initiatives must be integrated with grassroot-level collective choice rules. This multi-level structure is necessary in such a poverty-stricken community, where government resources can be leveraged to support local-level resource management.

Page 12 line 389 "Institute SANB. 2016 Classification system for wetlands and other aquatic ecosystems. Pretoria, South Africa: Government Printer." I suspect that this reference has been incorrectly cited and

should appear as follows: “Ollis DJ, Snaddon CD, Job NM, Mbona N 2013. Classification System for Wetlands and other Aquatic Ecosystems in South Africa. User Manual: Inland Systems No. 22, SANBI Biodiversity Series. South African National Biodiversity Institute, Pretoria. Draft final report to the Water Research Commission, Pretoria.”

***** Reply 24:** This was indeed a error. We have made this correction (reference 22)